# Code Aesthetics with Agentic Reward Feedback

**Bang Xiao**[1,2*]   **Lingjie Jiang**[1,3*]   **Shaohan Huang**[1†]   **Tengchao Lv**[1]
**Yupan Huang**[1]   **Xun Wu**[1]   **Lei Cui**[1]   **Furu Wei**[1]

[1] Microsoft Research Asia
[2] Zhiyuan College, Shanghai Jiao Tong University
[3] Peking University

sherlock_bang@sjtu.edu.cn   lingjiejiang@stu.pku.edu.cn
{shaohanh, fuwei}@microsoft.com

## Abstract

Large Language Models (LLMs) have become valuable assistants for developers in code-related tasks. While LLMs excel at traditional programming tasks such as code generation and bug fixing, they struggle with visually-oriented coding tasks, often producing suboptimal aesthetics. In this paper, we introduce a new pipeline to enhance the aesthetic quality of LLM-generated code. We first construct AesCode-358K, a large-scale instruction-tuning dataset focused on code aesthetics. Next, we propose *agentic reward feedback*, a multi-agent system that evaluates executability, static aesthetics, and interactive aesthetics. Building on this, we develop GRPO-AR, which integrates these signals into the GRPO algorithm for joint optimization of functionality and code aesthetics. Finally, we develop *OpenDesign*, a benchmark for assessing code aesthetics. Experimental results show that combining supervised fine-tuning on AesCode-358K with reinforcement learning using agentic reward feedback significantly improves performance on *OpenDesign* and enhances results on existing benchmarks such as *PandasPlotBench*. Notably, our AesCoder-4B surpasses GPT-4o and GPT-4.1, and achieves performance comparable to large open-source models with 480B–685B parameters, underscoring the effectiveness of our approach.

## 1 Introduction

LLMs have become powerful assistants in our daily lives, helping us polish writing, refine code, and access knowledge (Team, 2025; DeepSeek-AI et al., 2025; OpenAI, 2025). Recently, coding LLMs have achieved great success in various code related fields, such as code completion, bug fixing, and software engineering(Anthropic, 2025; Guo et al., 2024). While LLMs have demonstrated remarkable capabilities in single-text-modality coding tasks, they remain inadequate in visually-oriented tasks such as *chart generation* and *webpage design*, leading to poor visual outcomes like overlapping elements, inconsistent color schemes, and disorganized structures. Consequently, the aesthetic dimension of LLMs remains an underexplored area.

In this paper, we focus on assessing and improving LLMs ability in **visually-oriented coding tasks**, which refer to programming tasks in which the correctness or quality of the code is inherently tied to its visual output. Typical examples include tasks that generate or manipulate visual artifacts such as web pages (HTML/CSS), plots and charts (e.g., Matplotlib (Hunter, 2007), Seaborn (Waskom, 2021), Plotly (Inc., 2015)), or graphical scenes (e.g., Python Turtle). Unlike purely algorithmic coding tasks, these tasks require the model to reason about visual structure, spatial layout, and aesthetic consistency, in addition to syntactic or functional correctness. For the visually-oriented coding tasks, a natural question arises: *do LLMs possess any awareness of the aesthetics of their own code?* In other words, *do they have a sense of aesthetics*?

---

*Equal contribution.   † Corresponding author.
Project pape: https://bangx7.github.io/code-aesthetics

Building on these insights, we propose the **code aesthetics** concept, which captures the aesthetic appeal of visually-oriented code. Currently, reward methods for training coding LLMs often focus on a single textual modality, such as code executability and result correctness (Gehring et al., 2024; Fu et al., 2023; Le et al., 2022; Dai et al., 2025). These methods have significant limitations when applied to code aesthetics tasks, as they fail to assess visual aesthetics and are unable to interact with elements like webpages, making them ineffective as reward sources. To address this challenge, we propose **agentic reward feedback**, a new reward framework consisting of three agents, (i) execution agent, which checks the code executability, (ii) static aesthetics agent, which assesses the aesthetics based on an image of code execution result, and (iii) interactive aesthetics agent, which is specified to evaluate the function of webpages while interacting with web elements. When receiving a raw model output, the execution agent will try to extract the code blocks from the output and check its executability. If passed, static aesthetics agent and interactive aesthetics agent will then run in parallel to assess the static and interactive aesthetics perspectives respectively. The core idea is simple: adopting a multi-agent system to provide a comprehensive and systematic reward feedback from textual, visual, and interactive perspectives, thus giving a comprehensive feedback to better align the sense of aesthetics of model with human or advanced models. This approach addresses a key limitation of most open-source coding LLMs, which are confined to a single textual modality and thus lack awareness of the visual rendering of their code.

To achieve this goal, we first build a large-scale supervised instruction tuning dataset **AesCode-358K** of two major code aesthetics tasks: Python-based plot generation and webpage design. Given the absence of existing benchmarks for evaluating webpage aesthetics, we also construct the **OpenDesign** benchmark, which consists of $840$ real webpage design cases, to evaluate the aesthetics of webpage from both visual (static) and interactive aesthetics using LLM-as-a-judge (Zheng et al., 2023; Gu et al., 2025) method. Consequently, we perform reinforcement learning using **GRPO** (Shao et al., 2024) algorithm combined with our **A**gentic **R**eward framework (**GRPO-AR**) to train two models with different parameter scales—AesCoder 4B and AesCoder 7B. After supervised fine-tuning on the AesCode-358K dataset and reinforcement learning with GRPO-AR, our models achieve significant improvement in PandasPlotBench(Galimzyanov et al., 2025) and OpenDesign, showcasing the effectiveness of the AesCode-358K dataset and GRPO-AR method.

The key contributions can be summarized as follows:

- We introduce the concept of **code aesthetics** and investigate whether LLM-generated code demonstrates its own design aesthetics.
- We construct the first dataset for code aesthetics, AesCode-358K, and introduce the first benchmark, **OpenDesign**, a benchmark specifically designed to assess webpage design aesthetics.
- We propose a novel reward framework for code aesthetics, **agentic reward feedback**, and combine it with GRPO algorithm for more effective model training in code aesthetics tasks.

## 2 RELATED WORKS

**Aesthetics of AI-Generated Contents.** With the rapid advancement of generative artificial intelligence (van der Zant et al., 2013; Sakirin & Kusuma, 2023; Jovanovic & Campbell, 2022), increasing attention has been directed to the aesthetic taste of AI-generated content (AIGC) (Cao et al., 2025; Wu et al., 2023) and the alignment between AI aesthetics and human preferences (Zhang et al., 2024; Liao et al., 2025; Ouyang et al., 2022). Previous works include textual aesthetics (Jiang et al., 2024; Dilley, 2016), which investigates methods to provides a cleaner layout and better coherence of LLM's output (Jiang et al., 2024), and image aesthetics (Deng et al., 2017; Wu et al., 2024), which focuses on assessing and improving the aesthetic quality of images. However, all these methods rely on evaluating static image(s) and may not capable to assess contents like webpages which need interactions. As the growing maturity of AI agents (Achiam et al., 2023; Hurst et al., 2024), it becomes possible to integrate interactive evaluation into the contents generated by large language models, thereby providing more comprehensive and systematic feedback.

**Reward Systems in Reinforcement Learning.** In reinforcement learning, the reward serves as a scalar feedback signal that quantitatively evaluates the immediate desirability of an agent's actions, thereby guiding the learning process toward behaviors that maximize cumulative long-term return (Kaelbling et al., 1996). In the context of training large language models, the sources of reward can

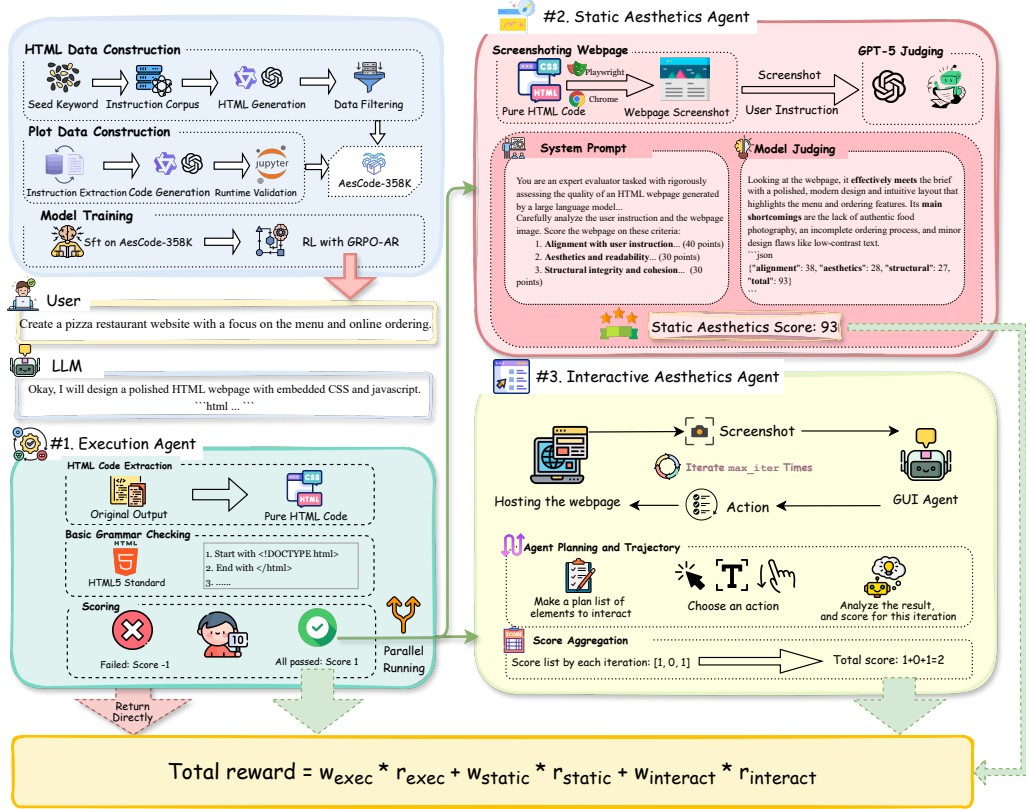

Figure 1: Overview of the AesCoder pipeline, which integrates data construction, model training, and a weighted scoring mechanism. GRPO-AR coordinates performing GRPO with three specialized reward agents—Execution, Static Aesthetics, and Interactive Aesthetics—for comprehensive reward feedback.

be broadly categorized into two main types: (i) Model-based Rewards: This approach utilizes a pretrained reward model to generate feedback (Ouyang et al., 2022; Christiano et al., 2017; Wang et al., 2024a; Cui et al., 2023). These models encode human preferences or expert knowledge, providing an automated and scalable source of reward. (ii) Rule-based Rewards: This type of reward is generated directly from human-defined rules or logic (Shao et al., 2024; Xie et al., 2025; Mu et al., 2024). However, in complex tasks, relying solely on a single source of reward can induce biased behaviors, ultimately driving optimization in an incorrect direction. Some works have been attempting to use agents, which combine human preference rewards with verifiable signals, to provide more reliable rewards (Peng et al., 2025).

## 3 THE AESCODE-358K DATASET

To investigate code aesthetics, we focus on domains where both the visual outcome and the implementation style matter. In this context, two representative areas are considered: *Python-based plot generation*, which emphasizes clarity and expressiveness in visualization, and *webpage design*, where aesthetic factors directly influence layout and user experience. In this section, we introduce **AesCode-358K**, a large-scale supervised instruction-tuning dataset designed for two key areas of code aesthetics.

### 3.1 PYTHON-BASED PLOT DATA CONSTRUCTION

We adapted instructions from the existing VisCode-200K dataset (Ni et al., 2025). While the original dataset contains 200K data points, we found that some of the Python code snippets were not

executable. To ensure high quality, we used Qwen3-Coder-480B-A35B-Instruct-FP8 (Team, 2025; Hui et al., 2024) to regenerate the Python code.

We enforced quality control in two ways. First, we limited the Python environment to essential libraries like `matplotlib`, `seaborn`, and `plotly` to prevent unexpected imports. Second, we validated the code's executability using Jupyter Notebook runtime checks, ensuring that the generated code runs without errors and produces the correct visualizations. After this rigorous filtering, we obtained 158K high-quality plot data points.

## 3.2 Webpage Design Data Construction

We developed a four-step process to create a large-scale webpage design dataset. First, we used GPT-4o to generate a seed keyword corpus across five webpage categories: *General Website*, *3D Design*, *Data Visualization*, *Game Dev*, and *UI Component*. Next, GPT-4o was used to produce diverse webpage design instructions from these keywords. We then projected the instructions into an embedding space and applied t-SNE visualization to examine category overlap. To remove redundancy, we further applied large-scale clustering and retained only representative samples, resulting in a refined instruction dataset (details in Appendix B.2). Finally, we employed GPT-5 (OpenAI, 2025) and Qwen3-Coder-480B-A35B-Instruct-FP8 (Team, 2025) to generate HTML code for each instruction. We present dataset statistics and keyword generation prompts in Appendix B.

To ensure the quality of the generated HTML code, we first confirmed that it was executable. We then rendered the webpages using `playwright` [1] and `selenium` [2] and asked GPT-5 to score the two outputs based on their rendered images. We selected the code with the higher score as our final data.

# 4 Agentic Reward Framework

For coding tasks, mainstream reward signals typically include execution or unit test success (Gehring et al., 2024; Fu et al., 2023), process-aware reward models (Le et al., 2022; Dai et al., 2025), and human preference feedback (Shen et al., 2023). However, these approaches mainly focus on textual modality and lack vision-oriented reward signals, rendering them unsuitable for evaluating code aesthetics. In visually grounded code generation, we highlight three essential dimensions:

- **Code Executability**. The generated code must run successfully, which forms the fundamental requirement of all code-related tasks.
- **Static Aesthetics**. This dimension captures the visual quality of the rendered output. An effective design should be concise, well-structured, and visually coherent, with elements properly aligned and exhibiting a clear sense of design.
- **Interactive Aesthetics**. Beyond static visuals, interactive aspects are crucial for webpages—especially those featuring 3D objects or browser-based games. This dimension evaluates whether page elements are not only visually appealing but also functionally meaningful and reasonably interactive.

Based on these dimensions, we propose an *agentic reward framework* that leverages a multi-agent system to assess each aspect, integrates their evaluations, and generates comprehensive feedback for webpage design from multiple perspectives.

## 4.1 Execution Agent

The execution agent verifies whether the model's output is executable and reports the result to the feedback system. Specifically, it assigns $s_{\mathrm{exec}} = 1$ if the output passes all validations, and $s_{\mathrm{exec}} = -1$ otherwise. For a raw model output, the agent first attempts to extract the HTML code from the html block; if not found, the entire output is treated as HTML. Given that web browsers tolerate many structural and syntactic errors, strict execution checking is unsuitable for HTML. Instead, we use

---

[1] `https://playwright.dev/`
[2] `https://www.selenium.dev/`

HTMLHint [3] to implement a rule-based HTML checker to validate the basic syntax. The detailed rules can be seen in Appendix G.7.

## 4.2 STATIC AESTHETICS AGENT

The static aesthetics agent evaluates visual quality using full-page webpage screenshots. For an HTML file, it first hosts the page locally using `playwright` in headless mode, then captures a full-page screenshot for subsequent visual assessment. We identify three dimensions essential for evaluating a webpage screenshot:

- **Instructional Alignment**. Evaluates consistency between the page's style and user instructions.
- **Visual Elements**. Assesses the effective use of modern design features such as lighting, transparency, and gradients.
- **Layout and Cohesion**. Examines whether the structure is functional, responsive, and visually coherent, with concise yet design-aware typography.

We select GPT-5 (OpenAI, 2025) as the judge for its strong multimodal reasoning ability. Given an HTML file generated by the model under a prompt, the page is first rendered into a static image. Using a chain-of-thought approach (Wei et al., 2023), the judge evaluates this image and provides both a score and a rationale for each dimension. While both scores and explanations are required to ensure reliable evaluation (Wei et al., 2023; Yao et al., 2023), we retain only the final aggregated score as the output of the static aesthetics agent. The detailed prompts are provided in Appendix G.2.

## 4.3 INTERACTIVE AESTHETICS AGENT

For webpage design, evaluation based only on static screenshots is insufficient, as it overemphasizes visual appearance while neglecting usability. This issue is particularly critical for interactive webpages such as 3D design platforms or browser-based games. To address this, we introduce the *interactive aesthetics agent*, which autonomously *navigates*, *explores*, and *interacts* with webpages to provide usability-aware feedback. Given the HTML code, the agent launches the page in a headless environment, interacts with its elements, and evaluates their functionality. We adopt WebVoyager (He et al., 2024) as the base framework and GPT-4o (OpenAI, 2024) as the multimodal model.

**Agent Planning.** At the start of evaluation, the agent generates an initial list of interaction candidates by reasoning about which elements are most relevant to the user instruction and webpage content. It then ranks these candidates and selects the top $N$ for execution. To ensure evaluations remain offline, interactions requiring internet access (e.g., social media logins) are excluded, focusing only on the core webpage functionality.

**Agent Interacting and Scoring.** The agent then executes the planned interactions step by step, recording whether each attempt succeeds or fails. After completing all interactions, it outputs a binary score list indicating success (1) or failure (0) for each action, and aggregates them into a final interaction score: $s_{\text{interact}} = \sum_{i=1}^{N} s_i$. This score is then returned to the agentic reward framework (see Appendix G.3 for the full prompt).

**Discussions.** Current web agents can handle most webpage operations (He et al., 2024), but may still struggle with certain corner cases, such as confusing webpage elements or being misled by irrelevant textual content (Cemri et al., 2025; Wang et al., 2024b). Such agent failures lead to a score of 0 in the corresponding iteration, since we assign a score of 1 only when the webpage responds correctly. This may cause the agent to make incorrect judgments, resulting in scores lower than the true values. On the other hand, agent failures also partially reveal non-standard or sub-optimal aspects of webpage design. Therefore, despite these limitations, using web agents as evaluators provides a reasonable proxy for assessing overall webpage aesthetics and interactivity.

---

[3] https://htmlhint.com/

## 4.4 REWARD AGGREGATION

The results from the three agents are integrated by the agentic reward framework, which jointly evaluates execution, static aesthetics, and interactive aesthetics to provide comprehensive feedback on each webpage. Let $r_{\text{exec}}$, $r_{\text{static}}$, and $r_{\text{interact}}$ denote the rewards from the respective agents. The overall reward is then computed as

$$r = w_{\text{exec}} \cdot r_{\text{exec}} + w_{\text{static}} \cdot r_{\text{static}} + w_{\text{interact}} \cdot r_{\text{interact}}, \tag{1}$$

where $w$ represents the weight assigned to each agent.

## 5 AESCODER TRAINING

### 5.1 STAGE I: SUPERVISED FINE-TUNING ON AESCODE-358K

We perform supervised fine-tuning on two different model with different parameter scales on our AesCode-358K dataset: Qwen3-4B-Instruct-2507 (Team, 2025) and Qwen2.5-Coder-7B-Instruct (Hui et al., 2024). This validates the generalizability of AesCode-358K dataset and establish a robust foundation for next stage reinforcement learning.

### 5.2 STAGE II: REINFORCEMENT LEARNING WITH AGENTIC REWARD FEEDBACK

After supervised fine-tuning in stage I, the model acquires substantial high-quality knowledge. However, the model at this stage still exhibits limited generalization beyond the training distribution (Chu et al., 2025), especially in webpage design tasks. This limitation highlights the necessity of reinforcement learning (RL), which allows the model to adapt more flexibly and robustly to diverse and unseen scenarios. Thus, we perform reinforcement learning using the **GRPO-AR** method, which integrates the **GRPO** (Shao et al., 2024) algorithm with our **A**gentic **R**eward framework to enhance the model's ability.

**Data Preparation for RL.** For avoiding overlap with the data in AesCode-358K, which the model has already "seen" in stage I, we pick 20K RL data from WebSight v0.2 dataset (Laurençon et al., 2024). However, the user instructions in WebSight v0.2 are not categorized, so we use the original user instructions as seeds and use GPT-4o (OpenAI, 2024) to rewrite the instructions for clearer semantic expression. Prompts refer to Appendix G.6.

**GRPO with Agentic Reward.** To generalize model's webpage design ability, we adopt our agentic reward system as a reliable and robust reward provider and perform reinforcement learning using GRPO (Shao et al., 2024) algorithm. We call this training method as **GRPO-AR**. For each prompt $p$ in our RL dataset $\mathcal{D}_{RL}$, GRPO-AR samples a group of outputs $\{o_1, o_2, \ldots, o_G\}$ from the old policy model $\pi_{\theta_{old}}$ and our agentic reward framework will give each output a total reward $r_i$ from execution, static aesthetics, and interactive aesthetics perspectives respectively, yielding $G$ rewards $\{r_1, r_2, \ldots, r_G\}$ respectively. The advantage $\hat{A}_{i,t}$ can be caculated as follows:

$$\hat{A}_{i,t} = \frac{r_i - \text{mean}(r)}{\text{std}(r)} \tag{2}$$

Accordingly, the policy model is optimized by maximizing the GRPO objective under our agentic reward framework (GRPO-AR):

$$\mathcal{J}_{\text{GRPO}}(\theta) = \mathbb{E}[p \sim \mathcal{D}_{RL}, \{o_i\}_{i=1}^{G} \sim \pi_{\theta_{\text{SFT}}}(O|p)]$$

$$\frac{1}{G}\sum_{i=1}^{G}\frac{1}{|o_i|}\sum_{t=1}^{|o_i|}\left\{\min\left[\frac{\pi_\theta(o_{i,t}|p,o_{i,<t})}{\pi_{\theta_{\text{SFT}}}(o_{i,t}|p,o_{i,<t})}\hat{A}_{i,t}, \text{clip}\left(\frac{\pi_\theta(o_{i,t}|p,o_{i,<t})}{\pi_{\theta_{\text{SFT}}}(o_{i,t}|p,o_{i,<t})}, 1-\epsilon, 1+\epsilon\right)\hat{A}_{i,t}\right] - \beta\mathbb{D}_{\text{KL}}\left[\pi_\theta||\pi_{\text{ref}}\right]\right\} \tag{3}$$

## 6 THE OPENDESIGN BENCHMARK

Design Arena[4] is a widely used platform for benchmarking web page design, supported by a community of hundreds of thousands of voters. It allows users to submit web page designs and receive

---

[4]https://designarena.ai/

community feedback through voting. While effective, this voting process is time-consuming and impractical for large-scale evaluation.

To address this limitation, we introduce the *OpenDesign Benchmark*, which enables efficient and automated assessment of web page designs using large language models. The benchmark evaluates both static and interactive aspects of design and includes 840 real-world web page cases. A detailed breakdown of categories and their case counts is provided in the Appendix C.

We assess model performance from two perspectives: static aesthetics and interactive aesthetics. **Static evaluation**: given a prompt, the HTML generated by a model is rendered into a static image. The prompt and the image are then assessed by the static aesthetics agent (see Sec. 4.2), which pro-

duces a static aesthetics score. **Interactive evaluation**: using the same prompt and HTML code, the interactive aesthetics agent (see Sec. 4.3) assigns an interactive aesthetics score. The final benchmark score for a model is obtained by averaging these results across all benchmark cases.

To evaluate the quality and reliability of the OpenDesign benchmark, we adopt two complementary perspectives: (1) ranking consistency between OpenDesign and Design Arena, and (2) alignment between LLM scoring and human preference.

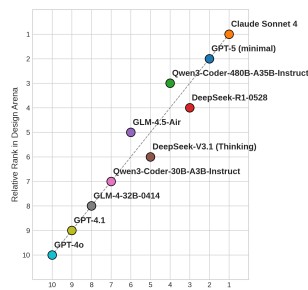 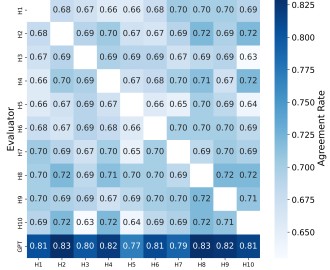

(a) Comparison of relative model rankings between Open-Design and Design Arena.

(b) Agreement rates among GPT and human evaluators. H-i refers to human evaluators.

To assess the reliability of our OpenDesign Benchmark, we compare the rankings of 10 mainstream foundation models against the Design Arena leaderboard[5]. We measure consistency using Spearman's and Kendall's rank correlation coefficients, obtaining strong agreement: Spearman = 0.98 ($p < 1.5 \times 10^{-6}$) and Kendall = 0.91 ($p < 3.0 \times 10^{-5}$). Additionally, OpenDesign achieves 66.7% top-3 and 80.0% top-5 overlap with Design Arena. These results indicate that OpenDesign closely reflects large-scale human judgment. Figure 2a plots model ranks across both benchmarks. Points align closely with the diagonal, confirming OpenDesign as a reliable proxy for human preferences in webpage aesthetics.

We sampled 200 HTML page pairs generated by the 10 models under the same prompts. Two evaluator groups—GPT judge and 10 humans (3 professors, 7 graduate students)—performed pairwise comparisons (win/tie/lose), yielding 2,000 annotations. Figure 2b shows agreement ratios: human-human = 68.7%, GPT-human = 80.9%. These are comparable to MT-Bench results (66% and 70%, respectively) (Bai et al., 2024; Zheng et al., 2023), supporting LLM-as-a-Judge as an effective, robust method for assessing code aesthetics.

# 7 EXPERIMENTS AND RESULTS

## 7.1 EXPERIMENTAL SETUP

We evaluate the model's plot generation using **PandasPlotBench** (Galimzyanov et al., 2025) with the `head` descriptor and `vis` mode. For each case, the model generates code from an instruction; executability is checked, and if an image is produced, it is compared to the ground truth. **GPT-4o** scores each case from 0 to 100. This results in three quantitative results, (i) error rate, which refers to the portion of cases do not pass the executability check, (ii) average score, which is the average GPT-4o score among all test cases, and (iii) good rate, which refers to the portion of scores higher than 75. Webpage design ability is assessed using our **OpenDesign** benchmark (see Section 6). Training settings are provided in the Appendix E.

---

[5]Rankings are taken as of September 22, 2025; Design Arena updates dynamically.

Table 1: Performance comparison between proprietary and open-source models across various benchmarks. In PandasPlotBench, Err., Avg., Good. refer to error rate, average score, good rate respectively. In OpenDesign, Align., Aes., Struct. refer to the three score perspectives: instructional alignment with user instruction, visual elements aesthetics, and structural cohesion respectively. Total. means the total score of the sum of three aspects' scores, and InterAes. refers to the score of interactive evaluation stage. Note: Lower is better for Err., higher is better for all other metrics. Best results are in **bold**, second-best results are underlined (among all open-source models together).

| Model | Size | PandasPlotBench | | | OpenDesign | | | | |
| | | Err. (↓) | Avg. (↑) | Good. (↑) | Static Aesthetics | | | | InterAes. (↑) |
| | | | | | Align. (↑) | Aes. (↑) | Struct. (↑) | Total. (↑) | |
| *Proprietary Models* | | | | | | | | | |
| GPT-4o-mini | - | 0.15 | 64 | 0.57 | 14.29 | 14.13 | 12.77 | 41.19 | 0.40 |
| GPT-4o | - | 0.09 | 68 | 0.60 | 16.90 | 16.05 | 15.13 | 48.08 | 0.44 |
| GPT-4.1 | - | 0.09 | 69 | 0.61 | 23.53 | 21.99 | 20.27 | 65.79 | 0.74 |
| GPT-5 (minimal) | - | 0.04 | 75 | 0.66 | 30.38 | 25.94 | 24.71 | 81.03 | **1.37** |
| Claude Sonnet 4 | - | 0.04 | 74 | 0.65 | 29.60 | 25.92 | 25.53 | 81.05 | 0.92 |
| *Open-Source Large Language Models* | | | | | | | | | |
| Qwen3-Coder-30B-A3B | 30B | 0.07 | 72 | 0.62 | 27.04 | 23.79 | 22.75 | 73.66 | 0.52 |
| GLM-4-32B-0414 | 32B | 0.07 | 70 | 0.59 | 24.67 | 22.90 | 21.80 | 69.40 | 0.48 |
| GLM-4.5-Air | 110B | 0.08 | 71 | 0.63 | 29.29 | 24.83 | 24.04 | 78.16 | 0.93 |
| Qwen3-Coder-480B-A35B | 480B | **0.05** | **73** | **0.66** | 30.13 | 25.16 | 24.62 | 79.90 | 0.70 |
| DeepSeek-V3.1 | 685B | 0.09 | 69 | 0.58 | 29.35 | 24.37 | 24.00 | 77.72 | 0.88 |
| DeepSeek-R1-0528 | 685B | 0.08 | 70 | 0.63 | 30.02 | 24.69 | 24.09 | 78.86 | 0.77 |
| *Open-Source Small Language Models* | | | | | | | | | |
| Qwen3-4B-Instruct-2507 | 4B | 0.13 | 65 | 0.55 | 27.52 | 23.01 | 22.73 | 73.26 | 0.67 |
| Qwen2.5-Coder-7B-Instruct | 7B | 0.22 | 60 | 0.50 | 16.38 | 15.13 | 14.73 | 46.27 | 0.38 |
| AesCoder-4B (Ours) | 4B | 0.09 | 70 | 0.63 | **30.42** | **26.19** | **25.31** | **81.92** | 1.04 |
| AesCoder-7B (Ours) | 7B | 0.09 | 67 | 0.57 | 30.03 | 25.98 | 25.18 | 81.23 | 0.94 |

## 7.2 MAIN RESULTS

As shown in Table 1, both **AesCoder-4B** and **AesCoder-7B** achieve consistent improvements over their respective baselines. On *PandasPlotBench*, they achieve lower error rates and higher reliability, indicating stronger capability in generating correct plotting code. On *OpenDesign*, AesCoder achieves substantial improvements in both **static aesthetics** (alignment, visual appeal, and structure) and **interactive aesthetics**, surpassing all other open-source models. In particular, AesCoder matches or outperforms models with 30B–685B parameters, establishing new state-of-the-art results among open-source systems.

When compared with proprietary models, **AesCoder-4B** not only surpasses **GPT-4o** and **GPT-4.1** on both *PandasPlotBench* and *OpenDesign*, but also delivers results competitive with substantially larger systems. Although **GPT-5** and **Claude Sonnet 4** still retain a slight overall advantage, our models achieve comparable scores across several aesthetic dimensions. These findings underscore the effectiveness of GRPO-AR, demonstrating that reinforcement learning with agentic reward feedback consistently enhances performance across different architectures and scales.

We further conducted human evaluation (Appendix F), and the results show that AesCoder-4B consistently outperforms strong open-source baselines (GLM-4-32B-0414 and Qwen3-Coder-30B-A3B-Instruct), which further validates our results.

## 7.3 ANALYSIS

**Generalization of agentic reward.** We further analyze the reward dynamics during reinforcement learning, as illustrated in Figure 3. Both Qwen2.5-Coder-7B-Instruct-SFT and Qwen3-4B-Instruct-2507-SFT exhibit steadily increasing reward scores with training steps. This consistent upward trend indicates that the agentic reward framework provides stable and informative feedback, enabling continuous improvement across different model families and sizes. The results highlight the robustness of the

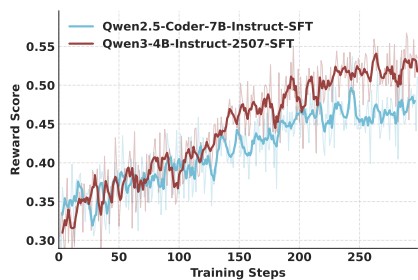

Figure 3: Reward curves during GRPO-AR.

framework as a general training signal, independent of specific architecture choices.

**Effect of Agentic Reward.** To isolate the contribution of the proposed agentic reward, we conduct a controlled comparison against a variant that does not incorporate it. Specifically, instead of leveraging the full agentic reward framework, we directly employ the underlying reward model to score model-generated HTML outputs along three static dimensions—Instructional Alignment, Visual Design and Aesthetics, and Structural Coherence and Usability—and use these scores as the sole reward signal (see Appendix G.4 for the exact prompt). The policy optimization strictly follows the same procedure as in §5.2, with the updates computed according to Eq. 3, thereby ensuring a fair comparison.

As reported in Table 2, this simplified variant consistently underperforms the full method that integrates agentic reward feedback. The performance gap highlights that merely reusing the reward model in a static fashion is insufficient. In contrast, our agentic reward framework, which incorporates multi-perspective evaluations including execution, static, and interactive aesthetics, provides richer and more reliable feedback. These results demonstrate that agentic reward is essential for aligning the model with both functional correctness and human-perceived aesthetics.

Table 2: Comparison with DPO, RFT, and ablations on Agentic Reward for Qwen3-4B-Instruct-2507 and Qwen2.5-Coder-7B-Instruct.

| Training Strategy | Align | Aes | Struct | InterAes |
|---|---|---|---|---|
| **Qwen3-4B-Instruct-2507** | | | | |
| Baseline | 28.50 | 25.27 | 24.36 | 0.62 |
| RFT | 29.32 | 25.30 | 24.67 | 0.71 |
| DPO | 28.79 | 25.31 | 24.38 | 0.70 |
| GRPO-AR w/o Agentic Reward (ablation) | 29.16 | 25.20 | 24.67 | 0.71 |
| **GRPO-AR w/ Agentic Reward (ours)** | **30.42** | **26.19** | **25.31** | **1.04** |
| **Qwen2.5-Coder-7B-Instruct** | | | | |
| Baseline | 28.85 | 25.23 | 24.37 | 0.70 |
| RFT | 29.73 | 25.35 | 24.85 | 0.75 |
| DPO | 29.75 | 25.33 | 24.87 | 0.71 |
| GRPO-AR w/o Agentic Reward (ablation) | 28.81 | 25.02 | 24.41 | 0.72 |
| **GRPO-AR w/ Agentic Reward (ours)** | **30.03** | **25.98** | **25.18** | **0.94** |

**Comparison with DPO and RFT.** To further validate the effectiveness of our proposed method GRPO-AR, we additionally compare it with two RLHF methods: Direct Preference Optimization (DPO) (Rafailov et al., 2024) and Rejection Sampling Fine-Tuning (RFT) (Yuan et al., 2023). Both methods are applied to the Stage I checkpoint $\pi_{\theta_{\text{SFT}}}$, using the same training data as in Stage II to ensure a fair comparison. Implementation details of DPO and RFT are provided in Appendix D. As shown in Table 2, our method consistently surpasses both DPO and RFT on OpenDesign across static and interactive aesthetics. These improvements highlight that incorporating agentic reward feedback not only enhances the visual quality of generated webpages but also strengthens their usability and structural robustness, confirming the superiority of GRPO-AR.

## 8 CASE STUDY

We further conduct case studies on the OpenDesign benchmark to qualitatively compare AesCoder-4B with Claude Sonnet 4 (Anthropic, 2025) and DeepSeek-R1-0528 (DeepSeek-AI et al., 2025). We select five representative cases from the five categories in OpenDesign for comparison. As illustrated in Figure 4, AesCoder-4B achieves results that are superior to or on par with state-of-the-art models across all five web design task categories. These results highlight the effectiveness of our approach in aligning code generation with both usability and aesthetic quality.

## 9 CONCLUSION

In this work, we introduce the concept of *code aesthetics* and present **AesCode-358K**, **OpenDesign**, and an **agentic reward framework (GRPO-AR)** that jointly enhance executability, static design, and interactivity in code generation. Through supervised tuning and reinforcement learning with GRPO-AR, our AesCoder models achieve state-of-the-art results on PandasPlotBench and Open-Design, rivaling much larger models. These results demonstrate that multi-agent reward feedback can effectively align coding LLMs with both functional correctness and human-perceived aesthetics, paving the way for more capable and user-friendly coding assistants.

| AesCoder-4B (Ours) | Claude Sonnet 4 | DeepSeek-R1-0528 |
| --- | --- | --- |

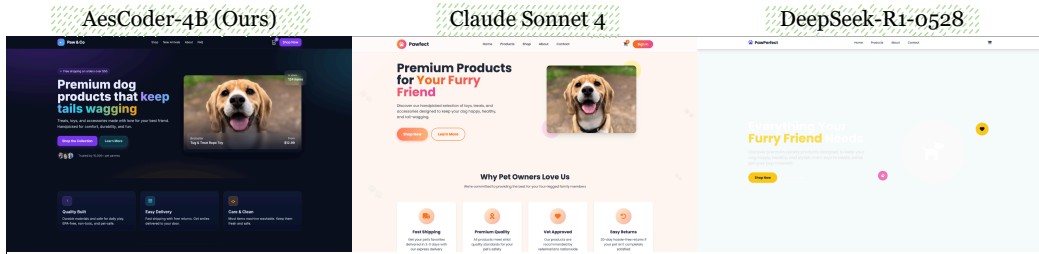

*Prompt: Create a user-friendly website for a landing page dedicated to selling dog-related products, ensuring easy navigation and an appealing design for pet owners.*

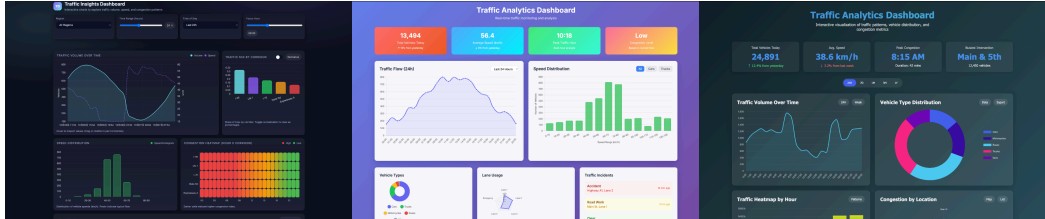

*Prompt: Create a template to display traffic data using interactive charts and graphs.*

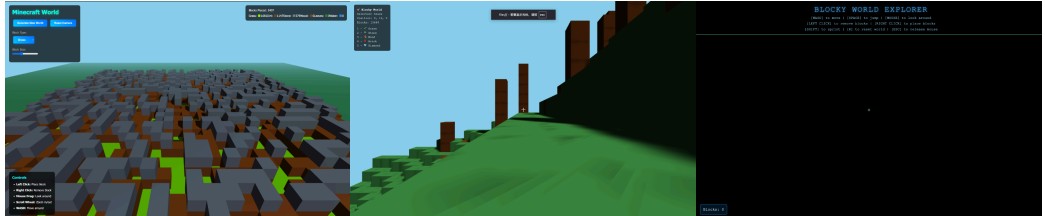

*Prompt: Create a blocky virtual landscape reminiscent of Minecraft, where players can explore and interact with a pixelated 3D world.*

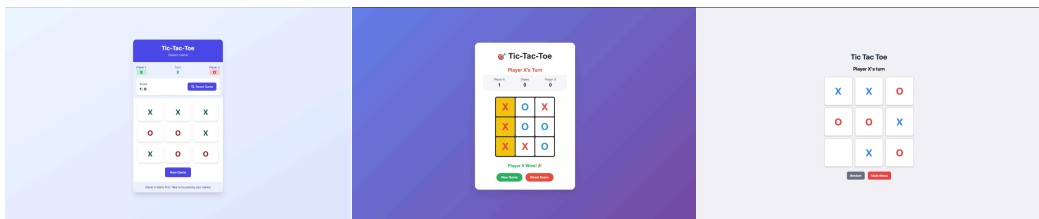

*Prompt: Create an interactive Tic-Tac-Toe game for the browser, allowing two players to take turns marking Xs and Os on a 3x3 grid.*

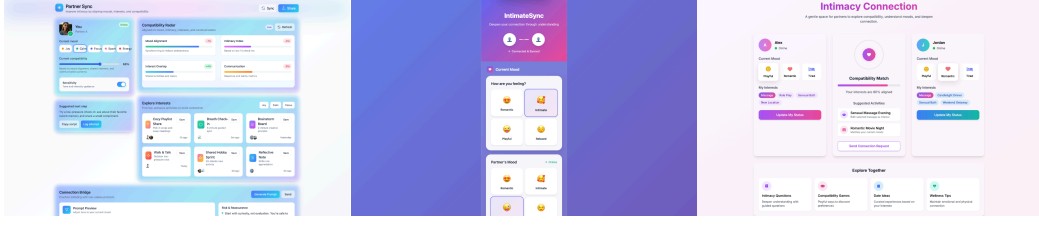

*Prompt: Create a user interface for partners aiming to improve sexual compatibility, explore interests, and understand each other's moods. This helps address issues like lack of intimacy and reduces awkwardness or rejection when initiating interactions.*

Figure 4: Case study comparing AesCoder-4B and baseline models on OpenDesign. The categories from top to bottom are: *General Website, Data Visualization, 3D Design, Game Dev, UI Component*.

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

## A    LLM Usage Statement

A large language model (ChatGPT) was used to **aid and polish the writing of the paper**, including minor grammar correction and language refinement.

## B    Details of Web Page Data Construction

### B.1    Keyword Corpus and Instruction Generation

We classified webpages into five categories: *General Website*, *3D Design*, *Data Visualization*, *Game Dev*, and *UI Component*. Using GPT-4o, we generated 9K seed keywords for the *General Website* category, and 2.5K keywords for each of the remaining four categories. Table 3 summarizes the distribution.

Table 3: Seed keywords statistics across categories.

| **Category** | General Website | 3D Design | Data Visualization | Game Dev | UI Component |
|---|---|---|---|---|---|
| **Seed Keywords** | 9,000 | 2,500 | 2,500 | 2,500 | 2,500 |

Based on the seed corpus, GPT-4o was asked to generate **20 non-redundant and semantically diverse instructions** for each keyword. This resulted in a total of 400,000 webpage design instructions for further processing.

### B.2    Semantic Analysis and Deduplication

We embedded all instructions using `openai-text-embedding-3-large` (3072 dimensions). From each category, 2,000 instructions were randomly sampled and visualized with t-SNE (perplexity = 30, max_iter = 1000). As shown in Figure 5, the raw dataset exhibited significant overlaps across categories, along with several dense clusters.

To filter out redundancy, we applied K-Means clustering with $K = 200$K on the embedded vectors and kept only the sample nearest to each cluster center. This resulted in a refined dataset of 200K instructions. The t-SNE visualization of the refined dataset shows clearer class boundaries and reduced overlap across categories, demonstrating the effectiveness of our filtering.

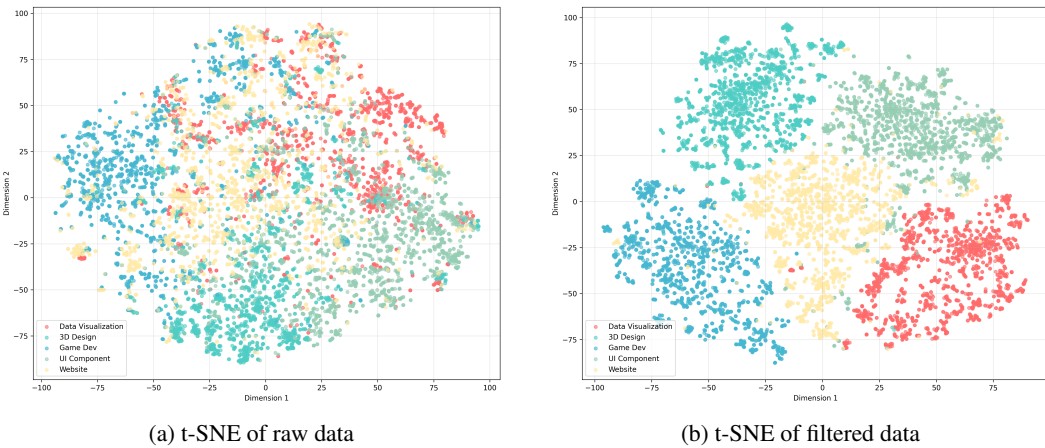

(a) t-SNE of raw data                    (b) t-SNE of filtered data

Figure 5: Visualization of instruction embeddings before and after filtering.

Table 4: Distribution of OpenDesign Benchmark Categories (Total: 840 cases)

| General Website | 3D Design | Data Visualization | Game Dev | UI Component | Total |
|---|---|---|---|---|---|
| 60.9% | 14.6% | 4.8% | 13.6% | 4.9% | 100% |

## C  OPENDESIGN BENCHMARK CATEGORIES

## D  IMPLEMENTATION DETAILS FOR DPO AND RFT

In this section, we describe the construction pipeline of training data for both DPO and RFT used in §7.3. We adopt the same set of queries as in GRPO-AR for offline sampling. For each query $q$, we sample $N$ responses from the SFT policy $\pi_{\theta_{\text{SFT}}}$, yielding

$$\mathcal{O}(q) = \left\{ o_i \right\}_{i=1}^{N}. \tag{4}$$

A reward model $R_\phi$ then scores each response conditioned on $q$:

$$\mathcal{R}(q) = \left\{ r(o_i \mid q) \mid o_i \in \mathcal{O}(q) \right\}, \quad \text{where } r(o \mid q) \equiv R_\phi(o \mid q). \tag{5}$$

**DPO.**  For DPO, we construct a preference dataset by taking, for each $q$, the highest- and lowest-scoring responses:

$$\mathcal{D}_{\text{DPO}} = \left\{ (q, o_w, o_l) \mid o_w = \arg \max_{o \in \mathcal{O}(q)} r(o \mid q),\ o_l = \arg \min_{o \in \mathcal{O}(q)} r(o \mid q) \right\}. \tag{6}$$

We then optimize $\pi_\theta$ (initialized from $\pi_{\theta_{\text{SFT}}}$) with the standard DPO objective (Rafailov et al., 2024):

$$\max_{\theta}\ \mathbb{E}_{(q, o_w, o_l) \sim \mathcal{D}_{\text{DPO}}} \left[ \log \sigma \left( \beta \left( \log \frac{\pi_\theta(o_w|q)}{\pi_{\theta_{\text{SFT}}}(o_w|q)} - \log \frac{\pi_\theta(o_l|q)}{\pi_{\theta_{\text{SFT}}}(o_l|q)} \right) \right) \right], \tag{7}$$

where $\sigma(\cdot)$ is the sigmoid and $\beta > 0$ is a scaling hyperparameter.

**RFT.**  For RFT, we select only the top-scoring response per query:

$$\mathcal{D}_{\text{RFT}} = \left\{ (q, o) \mid o = \arg \max_{o \in \mathcal{O}(q)} r(o \mid q) \right\}. \tag{8}$$

The model is then trained with a standard supervised objective:

$$\mathcal{L}_{\text{RFT}}(\theta) = - \mathbb{E}_{(q,o) \sim \mathcal{D}_{\text{RFT}}} \left[ \sum_{t=1}^{|o|} \log \pi_\theta(o_t \mid q, o_{1:t-1}) \right]. \tag{9}$$

**Implementation.**  We implement both DPO and RFT with LLaMA-Factory (Zheng et al., 2024)[6]. For a fair comparison with GRPO-AR, we keep the same learning rate, batch size, and the total number of training samples as in Stage II.

## E  TRAINING SETTINGS.

For stage I, all models are trained for 3 epochs with the AdamW optimizer, employing a 10% linear warmup followed by a cosine learning rate decay schedule. The maximum learning rate is set to $1e-5$, with a batch size of $128$ and a maximum sequence length of 8k tokens. Training the 7B model in the SFT phase takes approximately 2 days on 1 nodes of 8xMI300 GPUs.

For stage II, we use VeRL (Sheng et al., 2024) to conduct experiments. By default, we use a constant $3 \times 10^{-6}$ learning rate together with AdamW optimizer for policy model, and use a batch size of 64 and micro batchsize of 8. The rollout stage collects 64 prompts and samples 8 responses for each prompt. We set KL coefficient to 0.001 and $\epsilon = 0.5$ in Eq. 3 in all experiments. The RL phase

---

[6]https://github.com/hiyouga/LLaMA-Factory

takes approximately 7 days on 1 nodes of 8xMI300 GPUs. In agentic reward framework, we set $w_{exec} = 0.1$, $w_{static} = 0.8$, and $w_{interact} = 0.1$. Given the currently low success rate of GUI agents (Zhang et al., 2025; Nguyen et al., 2025; He et al., 2024), we limit the number of interactive elements to 3 during training. Additionally, when the GUI agent lists the interactive elements, we instruct it to prioritize them based on their importance. This ensures that the most critical and prominent elements are interacted with, thereby mitigating the impact of the GUI agent's limited success rate on our GRPO-AR training.

## F    HUMAN EVALUATION

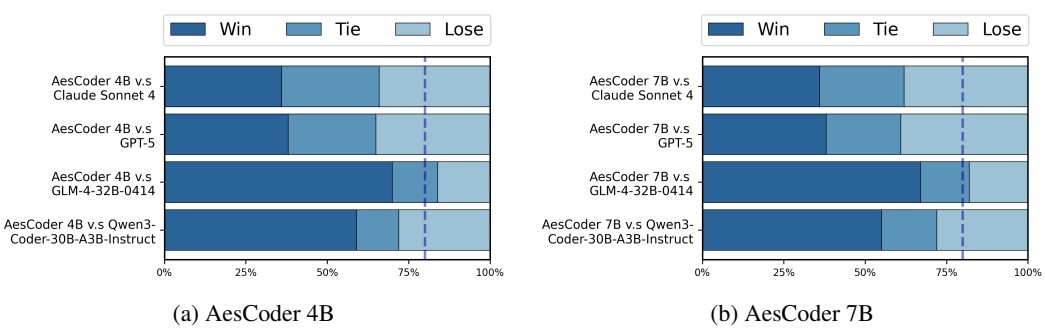

(a) AesCoder 4B                    (b) AesCoder 7B

Table 5: Human preference result visualization of AesCoder and other models.

To validate the effectiveness of our model, we select four mainstream models, Claude Sonnet 4 (Anthropic, 2025), GPT-5 (OpenAI, 2025), GLM-4-32B-0414(GLM et al., 2024) and Qwen3-Coder-30B-A3B-Instruct (Team, 2025) and randomly sampled 100 test cases from OpenDesign, resulting in 100 HTML pairs $\langle \pi_{ours}(p), \pi_{others}(p) \rangle$. Then we perform the same human preference annotations as Section 6. Results are shown in Figure 5. AesCoder achieves a win rate of over 55% in comparisons with mid- to large-scale open-source models (GLM-4-32B-0414 and Qwen3-Coder-30B-A3B-Instruct), and maintains a near 50% win rate when compared to state-of-the-art proprietary models (Claude Sonnet 4 and GPT-5), demonstrating the effectiveness of our agentic reward framework.

## G  PROMPT

### G.1  PROMPT TEMPLATE FOR PAIRWISE EVALUATION

---

**Prompt Template for pairwise Evaluation**

You are a highly-skilled and impartial AI evaluator. Your task is to distinctively evaluate two HTML webpage images, Image A and Image B, generated from the same user instruction but by different models. Your evaluation should emphasize clear differentiation and ranking between the two images, avoiding similar or average scores unless they are truly of equal quality. Always highlight meaningful differences.

You will be provided with the following:
- The general topic of the generated webpages: {topic}
- The original user instruction used to generate the webpages: {user_instruction}
- Image A, representing the output of the first model, which will be given later.
- Image B, representing the output of the second model, which will be given later.

Scoring Criteria (Total: 100 points per image):

1. Alignment with User Instruction (40 points):
- Score how well each image aligns with the details and intent of the provided user instruction.
- Assess whether all requested elements, content, and functionalities are present and correctly implemented.
- Evaluate if the overall structure and layout match the user's requirements.

2. Aesthetics and Readability (30 points):
- Score the visual appeal, design quality, and overall polish of each webpage.
- Assess factors like color scheme, typography, use of whitespace, and visual hierarchy.
- Evaluate the ease of reading and understanding the content. Is the text clear? Are the sections well-defined?

3. Structural Integrity and Responsiveness (30 points):
- Score the logical organization and structure of the webpage.
- Assess the overall layout and how the different components are arranged.
- Evaluate how well the design would adapt to different screen sizes (e.g., mobile, tablet, desktop), based on visual cues in the image.

Scoring Instructions:
- Distinctiveness is required: Avoid giving similar or average scores to both images unless they truly have no meaningful difference.
- Justify both high and low scores: If one image is clearly better in any aspect, assign a noticeably higher score.
- If an image has major flaws, do not hesitate to give a low score for that criterion.
- Do not use safe scores. Use the full range of the scoring scale if appropriate.

Your output must contain specific scores for each criterion of the two images, and the overall comparison symbol. The template of the output should strictly obey the following json format (alignment_score, aesthetics_score, structure_score are just the abbreviation of Alignment with User Instruction score, Aesthetics and Readability score, and Structural Integrity and Responsiveness score):

---

```
{
  "Image A Score": {
    "alignment_score": score_A_1,
    "aesthetics_score": score_A_2,
    "structure_score": score_A_3,
    "Total Score": total_score_A
  },
  "Image B Score": {
    "alignment_score": score_B_1,
    "aesthetics_score": score_B_2,
    "structure_score": score_B_3,
    "Total Score": total_score_B
  },
  "Overall Comparison": "comparison_symbol"
  "feedback": "feedback"
}
```

Where:
- For scores:
- score_A_1, score_A_2, score_A_3 are the scores for Image A in each category.
- score_B_1, score_B_2, score_B_3 are the scores for Image B in each category.
- total_score_A and total_score_B are the sum of the individual scores for each image.

- For comparison symbol:
- If Image A is far superior to B, the comparison symbol should be [[A¿¿B]].
- If Image A is better than B, the comparison symbol should be [[A¿B]].
- If Image A and B are of equal quality, the comparison symbol should be [[A=B]].
- If Image A is worse than B, the comparison symbol should be [[A¡B]].
- If Image A is far inferior to B, the comparison symbol should be [[A¡¡B]].

- For feedback:
- A concise summary (about 50 words) of your evaluation, explaining the strengths and weaknesses of the webpage in relation to the scores you've given.

## G.2 PROMPT TEMPLATE FOR POINTWISE EVALUATION

### Prompt Template for Pointwise Evaluation

You are an expert evaluator tasked with rigorously assessing the quality of an HTML webpage generated by a large language model. You will be given an image of the rendered HTML webpage and the original user instruction.

Your primary goal is to provide an objective, accurate, and discriminative score, using the full range of the scoring scale (0–100). Do not hesitate to give low or moderate scores if the webpage is average or has flaws. Only award high scores to webpages that are truly exceptional and nearly flawless according to professional standards.

You will be provided with:
- The general topic of the generated webpage: {topic}
- The original user instruction: {user_instruction}
- Image A, representing the output of the model to evaluate

Evaluation Instructions:
1. Carefully analyze the user instruction and the webpage image.
2. Score the webpage on the following criteria (use the full scoring range):

Alignment with User Instruction (40 points):
- Does the webpage fully and precisely satisfy all explicit and implicit requirements of the user's prompt?
- Are all requested elements present and correctly implemented?
- Does the content and structure directly correspond to the instruction?

Aesthetics and Readability (30 points):
- Is the webpage visually appealing, modern, and professionally designed?
- Are color, font, and spacing choices effective and consistent?
- Is the text easy to read and the layout clear?

Structural Integrity and Cohesion (30 points):
- Is the structure logical, well-organized, and cohesive?
- Do all sections flow smoothly and intuitively?
- Is the user experience (based on the image) seamless and easy to follow?

Scoring Principles (Read Carefully):
- Use the full range for each criterion (e.g., 0–40, 0–30). Average or flawed webpages should receive average or below-average scores.
- High scores (top 20% of each range) should be awarded only for work that meets or exceeds professional standards with virtually no flaws.
- If the webpage is missing elements, has visual issues, or organizational problems, score accordingly low.
- Provide a brief justification for any high or low score.

Score Interpretation Reference:
- 90–100: Outstanding, professional, nearly perfect.
- 70–89: Good but with noticeable issues or minor flaws.
- 50–69: Average, with clear limitations or several weaknesses.
- 30–49: Below average, significant flaws or missing requirements.
- 0–29: Poor, major requirements missing, very low quality.

Provide your final output in the following JSON format:

```
{
  "alignment_score": <score out of 40>,
  "aesthetics_score": <score out of 30>,
  "structure_score": <score out of 30>,
  "total_score": <sum out of 100>,
  "feedback": "<concise summary (about 30 words) explaining the
      strengths and weaknesses and justifying the scores>"
}
```

Remember: As an expert evaluator, do not inflate scores. Always judge by high professional standards and make full use of the scoring scale.

## G.3 PROMPT TEMPLATE FOR INTERACTIVE AESTHETICS AGENT

---

### Prompt Template for Interactive Aesthetics Agent

Imagine you are a distinguished website design judger. Now you are given a task about evaluating the practicality and aesthetic about the interactivity of a webpage. The webpages you are given are all single-paged, offline html files. User will later provide you with the specific topic (Only in these five topics: ["General website", "Game dev", "Data visualization", "3D design", "UI component"]) and the detailed description of this webpage. You should evaluate the webpage's interactivity and aesthetic based on the topic and the detailed description.

When evaluating the aesthetic of interactivity of a webpage, you should consider the following aspects:
- First, think thoroughly about all the ways of interactions with the webpage based on the topic, the detailed description given by the user and the webpage screenshot. Output your planned interations at the beginning of the task in your thought.
- Then, evaluate the interactivity of the webpage in order according to your planned interations. For each time of interaction, carefully compare the webpage before and after the interaction. The webpage should change according to the interaction. If the webpage is not changed or the change is not expected, it should not be considered as a good webpage.
- Since the webpage is offline, we do not expect changes which need internet connection. Specially, for textbox, you should plan both typing in the textbox and clicking the search button. It cannot be considered as a successful interation if only you successfully type in the textbox, but the webpage has not changed at all after clicking the search button.
- When your interaction does produce feedback, you still need to carefully consider whether that feedback is correct and logical. For example, if you click on a list and it merely displays the list, but clicking on an item within the list does not trigger any response, then no points should be awarded. Only correct feedback can earn points.
- Sometimes when you click a navigation button, the webpage will not change simply because it is already in the page you want to go. You should try to click another navigation button and click back again to check the interactivity of this navigation button.
- {GAME_EXTRA_PROMPT}

In each iteration, you will receive an Observation that includes a screenshot of a webpage and some texts. This screenshot will feature Numerical Labels placed in the TOP LEFT corner of each Web Element. Carefully analyze the visual information to identify the Numerical Label corresponding to the Web Element that requires interaction, then follow the guidelines and choose one of the following actions:
1. Click a Web Element.
2. Delete existing content in a textbox and then type content.
3. Wait. Typically used to wait for unfinished webpage processes, with a duration of 1 seconds.
4. Press the up arrow key. (Only can be used when the topic of the webpage is game dev)
5. Press the down arrow key. (Only can be used when the topic of the webpage is game dev)
6. Press the left arrow key. (Only can be used when the topic of the webpage is game dev)
7. Press the right arrow key. (Only can be used when the topic of the webpage is game dev)
8. FINISH. This action should only be chosen when all evaluations in your plan list have been finished.

Correspondingly, Action should strictly follow the format:
- Click [Numerical_Label]
- Type [Numerical_Label]; [Content]
- Wait
- UP
- DOWN
- LEFT
- RIGHT

- FINISH

Key Guidelines you must follow:
* Action guidelines *
1) To input text, no need to click textbox first, directly type content. After typing, the system automatically hits ENTER key. Sometimes you should click the search button to apply search filters. Try to use simple language when searching.
2) If you have seen a scrollbar in the webpage (not for the whole window, since the webpage is always single-paged, but for a certain area or element of the webpage, such as a 3D object to be rotated or zoomed), do not directly try to scroll it. Instead, find if any interactable element such as button '-' or '+' and click the button instead.
3) If you click a button and then a pop-up window is displayed, you should close the pop-up window and return to the original webpage after you have finished evaluating the interaction.
4) If the topic of the webpage is game dev, it may not have many interactable elements to click. Instead, you can use the up, down, left, right arrow keys to control the game, and plan dynamically when the game running. Don't miss up the role in the game with interactable elements.
5) You must distinguish between textbox and search button, don't type content into the button. If no textbox is found, you may need to click the search button first before the textbox is displayed.
6) Execute only one action per iteration.
7) Strictly avoid repeating the same action if the webpage remains unchanged. You may have selected the wrong web element or numerical label. Continuous use of the Wait is also not allowed.
8) When a complex Task involves multiple questions or steps, select FINISH only at the very end, after addressing all of your planned interations. Flexibly combine your own abilities with the information in the webpage.

* Web Browsing Guidelines *
1) Don't try to go to other urls. Just focus on the given offline html page. All your interations can be done offline (without internet connection).
2) Focus on the numerical labels in the TOP LEFT corner of each rectangle (element). Ensure you don't mix them up with other numbers (e.g. Calendar) on the page.

Your reply should strictly follow the format:
For the first iteration (the planning stage):
Thought: {Your thorough plan to interact with all the interactable elements of the webpage}

For the other iterations (the interaction stage):
Thought: {Your brief thoughts (briefly summarize the info that will help you score the previous interaction, and your brief plan for the next interaction)}
Numerical_Label: {The numerical label of the previous interaction}
Score: {The score of the previous interaction. Only 0, 1, NaN is allowed. 0 means the interaction is failed or incorrect, 1 means successful. Output NaN if no interation is done in this iteration. Specially for textbox, you should output NaN when you finished typing in the textbox, and the actual score when you clicked the search button or something else.}
Reasoning: {Your brief reasoning for the score. Similarly, you must output N/A if no interation is done in the previous iteration}
Action: {One Action format you choose for the next interaction}

Then the User will provide:
Observation: {A labeled screenshot Given by User}

## G.4 PROMPT TEMPLATE FOR ABLATION WITHOUT AGENTIC REWARD

---

**Prompt Template for Ablation without Agentic Reward**

You are an expert evaluator tasked with assessing the quality of an HTML webpage generated by a large language model. You will be given the HTML code of the webpage and the original user instructions.

You will be provided with:
- The general topic of the generated webpage: {topic}
- The original user instruction: {user_instruction}
- The html code of the webpage: {html}

Your objective is to assign precise, rigorous scores, using the full 0–100 range. Only award high scores for webpages that are absolutely flawless, meeting all design and functional expectations. Penalize harshly for even the smallest imperfections—there is zero tolerance for errors.

Key Evaluation Areas:

1. Instructional Alignment (20 points)
Evaluate how closely the webpage follows the user's instructions. Only this in aspect, your criteria can be relatively low, since we expect some flexibility in interpretation and should more pay more attention in another two aspects (Visual Design and Aesthetics and Structural Coherence and Usability below).

Score levels and their explanations:
- Good alignment (10–20): The webpage almost matches the user's instructions.
- Severe misalignment (0–9): The page fails to meet basic requirements. Major elements are missing or misrepresented.

2. Visual Design and Aesthetics (50 points)
Assess the overall professionalism and polish of the design. Only award high marks for designs that look flawless, balanced, and intentional.

Some golden rules you should obey when scoring:
- Always cherish detailed, refined, and innovative design. A highly refined design is always better than a plain one, which means we value pages with highly rich design elements more than simple and plain designs. This includes an exquisite transparent dynamic background, elements or special effects floating in the background, gradient color text, rich yet beautiful color matching, and so on.
- NO PLACEHOLDERS! Always cherish real images and expressive (real or abstract) icons, instead of placeholders. A website with rich, real, and appropriate images or icons should score higher(85 or above), while a website with placeholders or broken images should score below 50. Abstract modern icon are also preferable, but they should be well-designed and are NOT placeholders.
- Simplicity is not a lack of content. A simple design can still be rich and engaging if it uses space, color, and typography effectively.
- The overall impression is important. Make sure the webpage has NO broken/partially visible words or elements. NO partially loaded elements.

Score levels and their explanations:
- Perfect design (40-50): The design is exceptionally professional, with a well-executed color palette, typography, and spacing. The page has a polished and intentional feel.
- Minor flaws (20-39): The design is good, but there are small issues (e.g., slight inconsistency in font sizes or spacing). These should still impact the score significantly.
- Significant flaws (10–19): The design has major issues (e.g., poor readability, awkward

---

layout, or jarring color choices).
- Unacceptable design (0–9): The page is unprofessional, with severe flaws such as overlapping text, unreadable fonts, or broken layouts / images.

3. Structural Coherence and Usability (30 points)
The page must have a logical and intuitive structure. Even the smallest structural mistake (misalignment, broken flow, or inconsistent layout) will severely affect the score.

Key scoring rules:
- Overall impression comes first. This stresses the importance of adopting a modern, concise, refined framework. Encourage websites to use modern, beautiful design frameworks instead of simple, mediocre designs. Webpages with appropriate use framework can score above 85, while those with poor or no framework should score below 50.
- Highlight the integrity of the overall structure. Check carefully whether the page has a complete structural layout, with no missing elements or broken sections. If the page has any broken sections, it should score below 50.

- Flawless structure (20–30): The page has a perfect structure: well-organized, logical flow, and easy navigation.
- Minor structural issues (15–19): The structure is good, but there are small usability issues (e.g., slightly misaligned sections or awkward navigation).
- Major structural problems (10–14): The page has significant usability flaws, such as broken layouts or confusing content organization.
- Unusable structure (0–9): The page has severe structural issues, making it difficult to use or navigate effectively.

Fine-Grained Scoring Guidelines:

- Strict threshold for high scores: Only give scores above 90 if the webpage is absolutely flawless. If there is even a minor issue (e.g., a single broken element, misalignment, or poorly chosen font), do not award high marks. Scores 95+ should be reserved for near perfection.
- Minor flaws are heavily penalized: If the webpage has any noticeable flaw (such as text overlapping an image, improper spacing, or a lack of balance), this will result in low overall score! (e.g., 10–30)
- Zero tolerance for bad design: If the webpage looks unprofessional (e.g., excessive white space, unaligned content/text, unreadable text, or poor contrast), the overall score should be 0-30!

Example Evaluation:

For a webpage with:
- Perfect alignment with instructions (everything is present and correct),
- Excellent visual design, but with slightly misaligned text,
- Clear structure with one misaligned image,

You might score:
- Instructional Alignment: 20/20 (perfect alignment with instructions),
- Visual Design: 35/50 (good design but minor flaw—misaligned text),
- Structural Coherence: 20/30 (minor misalignment of an image),
- Total Score: 75/100 (not good, but OK).

Final Output Format (alignment_score, aesthetic_score, structure_score are just the abbreviation of Instructional Alignment score, Visual Design and Aesthetics score, and Structural Coherence and Usability score):

```
{
  "alignment_score": <score out of 20>,
  "aesthetics_score": <score out of 50>,
  "structure_score": <score out of 30>,
  "total_score": <sum out of 100>,
  "feedback": "<brief summary of strengths and weaknesses, with
      justification for the scores>"
}
```

Strict Scoring Principles:
- Minor mistakes are penalized severely. A single misplaced element, broken layout, or poor design choice will dramatically affect the score.
- High scores (90+) should only be given for perfect webpages with no errors. If there is any imperfection, the score should drop significantly.
- No mercy for bad design. Webpages that are visually unappealing or hard to use must receive low scores (0–9) regardless of other factors.

## G.5 PROMPT FOR KEYWORD

---

**Prompt Template for Website Keyword and Summary Generation**

You are a professional website content generator. Generate 300 unique keywords or short summary descriptions (10–30 words each) for websites of the type "{catagory}". Each summary should:
- Reflect a unique purpose, functionality, or use case for a website.
- Be based on a creatively chosen theme or industry, covering a wide range of domains (e.g., healthcare, education, finance, entertainment, environmental, e-commerce, tourism, tech, art, sports, social impact, etc.) by leveraging your imagination.
- Ensure summaries are specific, diverse, and avoid repetition in functionality, theme, or wording.

Output as a JSON array, where each entry contains:
- summary: A concise description (10–30 words) of the website's purpose or functionality, reflecting the chosen theme.

Ensure maximum diversity by exploring unique and imaginative themes, avoiding overlap with common website concepts. Return the result in JSON format.

Example output format:

```
[
  {
    "summary": "A website for eco-conscious travelers, offering
        sustainable tourism guides, ethical lodging options, and
        carbon footprint calculators."
  },
  {
    "summary": "An educational platform providing interactive
        biology simulations, 3D models, and real-time quizzes for
        high school students."
  }
]
```

---

### G.6 Prompt Template for Data Rewriting in RL

---

**Prompt Template for Data Rewriting in RL**

You are a content strategist and creative visionary specializing in conceptualizing innovative digital platforms. Your task is to transform abstract ideas into compelling website concepts that are both unique and inspiring.

I will provide you with a brief description or seed topic. From this, your goal is to generate a highly imaginative and detailed website concept. The concept does not need to be directly correlated to the content I provide. Feel free to draw inspiration from keywords or abstract elements and create something new and innovative.

Your output should focus on the overall content, purpose, and features of the website, without going into specific layout, design, or visual details. Think about the theme, functionality, and interaction possibilities of the site. This will serve as the basis for generating HTML code for the site.

It has to be noticed that your instruction should not contain any specific layout, design, or visual elements of the website, but only the content, purpose, and features of the website.

You are required to choose one of the following categories for each website concept you create. Please try to think creatively and step outside the "General website" category when possible:

1. General website: A website designed for general use or any topic, focusing on its core content, purpose, and user interaction.
2. Game development: A browser-based game concept in HTML. Focus on interactive and engaging content, game mechanics, and user experience.
3. Data visualization: A page that presents dynamic and interactive data, such as charts, graphs, or visualized datasets. Focus on how the user will interact with and explore the data.
4. UI component: A page dedicated to showcasing a single, highly interactive component. Focus on the functionality and purpose of the component, without detailing its visual structure.
5. 3D design: A concept for a 3D scene or interactive experience, focusing on its content and user interaction, rather than specific rendering or layout details.

For each brief description I provide, follow this structure:
1. Select a category from the list above that best fits the concept.
2. Create a detailed and concise description of the website concept, focusing on its content, purpose, features, and interactions.
3. Provide a clear instruction (40–60 words) for HTML code generation that can be used to implement this concept.

Output the response in the following JSON format:

```
{
  "category": "<category name from the list>",
  "instruction": "<detailed website concept instruction>"
}
```

---

## G.7   EXECUTION AGENT VALIDATION RULES

### Execution Agent Validation Rules

The following configuration defines validation and linting rules for HTML, CSS, and JavaScript within a single HTML file. These rules should be strictly applied when evaluating or generating webpages.

```
{
  "doctype-html5": true,                    // Enforce HTML5
      doctype declaration
  "tagname-lowercase": true,                // Enforce lowercase
       tag names
  "attr-lowercase": true,                   // Enforce lowercase
       attribute names
  "attr-value-double-quotes": true,         // Enforce double
      quotes for attribute values
  "tag-pair": true,                         // Enforce all tags
      must have a corresponding closing tag
  "tag-self-close": ["br", "img", "input", "link", "meta"], //
      Allow self-closing tags for specific elements
  "id-unique": true,                        // Ensure 'id'
      attribute is unique in the document
  "alt-require": true,                      // Enforce 'alt'
      attribute for all  tags for accessibility
  "head-script-disabled": false,           // Allow <script>
      tags in the <head> section
  "style-disabled": false,                  // Allow inline CSS
      styles within HTML
  "no-inline-style": false,                 // Allow inline
      styles within HTML
  "no-inline-script": false,                // Allow inline
      JavaScript within the HTML file
  "lang-require": true,                     // Enforce 'lang'
      attribute in the <html> tag
  "meta-charset-utf-8": true,               // Ensure UTF-8
      charset declaration
  "meta-viewport": true,                    // Enforce inclusion
       of the viewport meta tag
  "title-require": true,                    // Enforce inclusion
       of the <title> tag
  "csslint": {
    "important": false,                     // Allow the use of
        !important in CSS
    "order-alphabetical": false             // Do not enforce
        alphabetical order for CSS properties
  },
  "script-disabled": false                  // Allow JavaScript
      (inline within HTML)
}
```

## G.8   PROMPT TEMPLATE FOR DATASET PROCESSING

### Prompt Template for Pointwise Evaluation

You are a professional evaluator tasked with performing a meticulous assessment of an HTML webpage generated by a large language model. You will receive both the rendered webpage image and the original user instruction.

Your goal is to assign precise, unbiased, and discriminative scores using a 0–100 scale. Use lower or mid-range scores for webpages that demonstrate average quality or contain flaws; reserve high scores exclusively for outputs that meet professional standards with minimal or no deficiencies.

You will be provided with:

- Webpage topic: **{topic}**
- Original user instruction: **{user_instruction}**
- Rendered webpage image: Image A

**Evaluation Procedure:**

1. Examine the user instruction carefully in conjunction with the webpage image.
2. Evaluate the webpage across the following criteria, making full use of the scoring ranges.

**1. Compliance with User Instruction (40 points)**

- Does the webpage satisfy all explicit and implicit requirements of the instruction?
- Are all requested elements present, correct, and properly implemented?
- Is the content and structure fully consistent with the user's instructions?

**2. Visual Design and Readability (30 points)**

- Is the webpage visually appealing, modern, and professionally designed?
- Are typography, color schemes, and spacing applied consistently and effectively?
- Is the text legible and the layout clean, clear, and easy to follow?

**3. Structural Cohesion and Organization (30 points)**

- Is the webpage structure logical, coherent, and well-organized?
- Do sections flow naturally and intuitively?
- Based on the image, is the user experience seamless and easy to navigate?

**Scoring Guidelines:**

- Use the full range of each criterion (0–40, 0–30, 0–30).
- Average or flawed webpages should receive average or below-average scores.
- High scores should be awarded only for outputs that meet professional standards with virtually no shortcomings.
- Deduct points for missing elements, visual issues, or structural inconsistencies.
- Provide a brief justification for unusually high or low scores.

**Score Interpretation:**

- 90–100: Outstanding; professional-quality; nearly flawless.
- 70–89: Strong; minor issues or noticeable imperfections.
- 50–69: Moderate; average quality with several limitations.
- 30–49: Weak; significant flaws or missing elements.
- 0–29: Poor; major requirements missing or very low overall quality.

**Output Format:** Provide your evaluation using the following JSON template:

```
{
  "alignment_score": <score out of 40>,
  "aesthetics_score": <score out of 30>,
  "structure_score": <score out of 30>,
  "total_score": <sum out of 100>,
  "feedback": "<concise summary (about 30 words) explaining the
      strengths and weaknesses and justifying the scores>"
}
```

**Important:** Maintain professional rigor. Avoid inflating scores and evaluate solely based on the quality observed in the rendered webpage image.

# H  ADDITIONAL EXPERIMENTS

## H.1  ROBUSTNESS OF OPENDESIGN EVALUATION UNDER ALTERNATIVE JUDGES

To assess whether the OpenDesign benchmark is robust to changes in the underlying aesthetic judge and to rule out circularity effects between training and evaluation, we additionally evaluate all mainstream foundation models using `Qwen3-VL-235B-A22B-Instruct` as an alternative judge. This model is a strong, open-source vision-language model independent of the OpenAI model family used in our main results.

Table 6 reports the results obtained by evaluating the same set of models with both GPT-5 and Qwen3-VL-235B-A22B-Instruct under the identical scoring prompt.

Table 6: OpenDesign evaluation with different judges. Despite differences in absolute score scales, the relative ranking of all models remains unchanged.

| Model Name | GPT-5 Eval | Qwen3-VL Eval | Relative Rank |
|---|---|---|---|
| AesCoder-4B (**Ours**) | 81.92 | 91.12 | 1 |
| Claude Sonnet 4 | 81.05 | 90.90 | 2 |
| GPT-5 (minimal) | 81.03 | 90.40 | 3 |
| Qwen3-Coder-480B-A35B | 79.90 | 90.37 | 4 |
| DeepSeek-R1-0528 | 78.86 | 89.50 | 5 |
| GLM-4.5-Air | 78.16 | 87.71 | 6 |
| DeepSeek-V3.1 (Thinking) | 77.72 | 87.62 | 7 |
| Qwen3-Coder-30B-A3B | 73.66 | 87.16 | 8 |
| GLM-4-32B-0414 | 69.40 | 81.41 | 9 |
| GPT-4.1 | 65.79 | 74.46 | 10 |
| GPT-4o | 48.13 | 70.52 | 11 |

The results show three key findings:

**1. OpenDesign evaluation is robust under judge substitution.**  Although Qwen3-VL-235B-A22B-Instruct assigns higher absolute scores than GPT-5, the *relative ordering* of the ten mainstream models remains *identical*. This consistency across two unrelated model families indicates that OpenDesign captures a judge-invariant notion of design quality rather than artifacts tied to a single evaluator.

**2. Improvements from AesCoder reflect genuine capability gains rather than judge-specific overfitting.**  Since Qwen3-VL-235B-A22B-Instruct and GPT-5 produce the same ranking across all baseline models, AesCoder's improvements cannot be attributed to exploiting idiosyncratic scoring patterns of a particular judge. Instead, the gains generalize across evaluators with different architectures, training data, and aesthetic priors, confirming that the improvements reflect true enhancement in visual coding ability.

**3. Both GPT-5 and Qwen3-VL-235B-A22B-Instruct exhibit alignment with human aesthetic preferences.**  Figure 2a in the main paper demonstrates strong diagonal agreement between GPT-5 and human pairwise judgments. Our new results show that Qwen3-VL-235B-A22B-Instruct replicates GPT-5's ranking structure on the same set of models. Since both judges independently recover the human-preferred ordering, this provides converging evidence that our evaluation methodology is scientifically well-grounded and reliably aligned with human aesthetic standards.

In summary, these experiments confirm that OpenDesign is robust to evaluator choice and that the improvements from AesCoder arise from real model capability rather than bias toward any particular judge model.

## H.2  SENSITIVITY ANALYSIS OF REWARD WEIGHT CONFIGURATION

To understand how different reward components influence learning dynamics in our agentic framework, we perform an ablation study by varying the weights $(w_{\text{exec}}, w_{\text{static}}, w_{\text{interact}})$ while keeping the

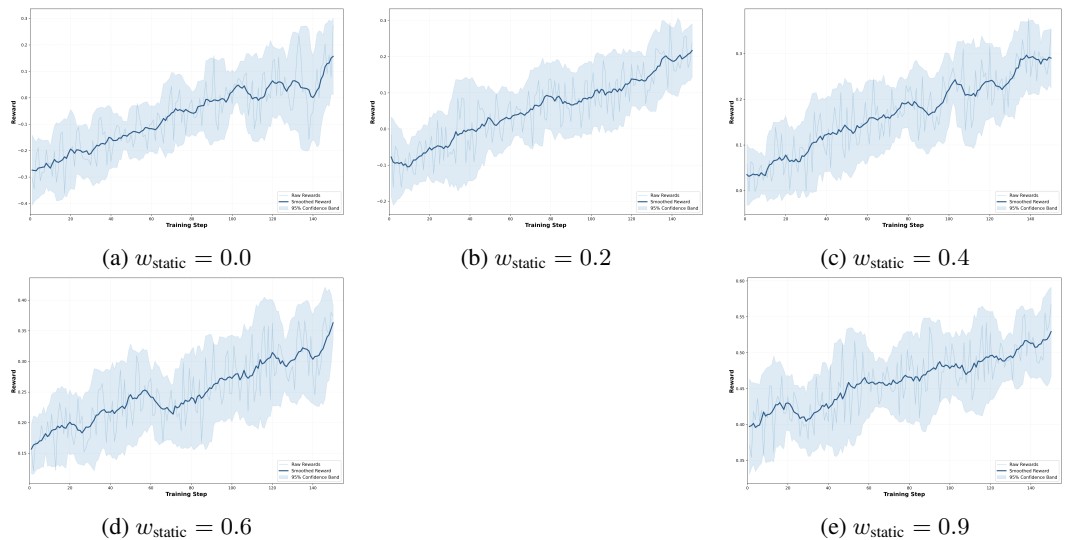

Figure 6: Reward curves for five weight configurations in GRPO-AR training.

total mass fixed. This analysis evaluates whether the framework is sensitive to weight perturbations and whether training remains stable under different reward trade-offs.

**Experimental Setup.** We keep $w_{\text{exec}} = 0.1$ fixed—reflecting the high tolerance of web browsers to HTML syntax—and redistribute the remaining mass between the Static Aesthetics and Interactive Aesthetics components. We select four representative configurations ranging from "pure static" to "pure interactive." For each configuration, we run GRPO-AR training for 150 RL steps on the AesCoder-4B$_{\text{sft}}$ model due to computational constraints. Each trained model is evaluated on the OpenDesign benchmark using 5 repeated runs to estimate confidence intervals.

Table 7: Effect of reward weight variation on OpenDesign performance. Mean $\pm$ std over 5 runs.

| $w_{\text{exec}}$ | $w_{\text{static}}$ | $w_{\text{interact}}$ | **Static Score** | **Interactive Score** |
|---|---|---|---|---|
| 0.1 | 0.0 | 0.9 | $78.88 \pm 0.32$ | $0.91 \pm 0.20$ |
| 0.1 | 0.2 | 0.7 | $79.28 \pm 0.30$ | $0.87 \pm 0.17$ |
| 0.1 | 0.4 | 0.5 | $79.87 \pm 0.33$ | $0.84 \pm 0.18$ |
| 0.1 | 0.6 | 0.3 | $80.01 \pm 0.33$ | $0.80 \pm 0.17$ |
| 0.1 | 0.9 | 0.0 | $80.50 \pm 0.32$ | $0.63 \pm 0.15$ |

**Findings.** The results reveal a clear and monotonic trade-off: increasing $w_{\text{static}}$ consistently improves static design quality, while decreasing interactivity; increasing $w_{\text{interact}}$ has the opposite effect. This smooth trend confirms that the agentic reward components are *well-behaved* and that the model optimizes rationally according to the specified weight structure.

Importantly, across all configurations tested, the training process remained stable and the reward curves (included in Figure 6) displayed normal upward trajectories without collapse. This demonstrates that GRPO-AR provides a controllable and robust optimization signal, and that the framework remains stable under substantial perturbations of reward composition.

**Conclusion.** These results indicate that (1) the model's behavior can be reliably steered by adjusting reward weights, (2) the system is not fragile to moderate weight changes, and (3) the reward components interact coherently within the GRPO-AR framework. Thus, the reward design is both flexible and stable, enabling practitioners to tune the model toward different aesthetic–functionality trade-offs depending on downstream needs.

### H.3 General Code Ability Evaluation on Standard Code Benchmarks

To examine whether our aesthetic-oriented reinforcement learning affects general-purpose coding ability, we evaluate both `AesCoder-4B` and `AesCoder-7B` on three standard benchmarks: **Live-CodeBench**(Jain et al., 2024), **MBPP** (Austin et al., 2021; Liu et al., 2023), and **MBPP+** (Austin et al., 2021; Liu et al., 2023). These benchmarks measure algorithmic correctness, functional reasoning, and code execution reliability—capabilities orthogonal to our aesthetic training targets.

Table 8: Results of AesCoder and their base models on general-purpose coding benchmarks.

| Model | LiveCodeBench | MBPP | MBPP+ |
|---|---|---|---|
| Qwen3-4B-Instruct-2507 | 32.5 | 86.8 | 74.3 |
| AesCoder-4B | 19.0 | 73.5 | 64.3 |
| Qwen2.5-Coder-7B-Instruct | 16.8 | 83.5 | 71.9 |
| AesCoder-7B | 15.3 | 73.5 | 63.1 |

Across all benchmarks, the AesCoder models exhibit **noticeable and expected regressions** in general code accuracy following aesthetic-focused reinforcement learning. This behavior is attributable to the alignment tax associated with domain specialization: GRPO-AR aggressively optimizes the model toward *code aesthetics* (e.g., Python-based plot generation and webpage design), diverging from the pure algorithmic reasoning objectives measured by standard benchmarks.

Crucially, while the performance drop is significant, **it reflects a conscious trade-off between becoming a domain expert in visually-oriented coding and maintaining state-of-the-art generalist capabilities**. The results indicate that the aesthetic alignment procedure shifts the model's distribution significantly toward design-oriented tasks.

In summary, this evaluation demonstrates that: (1) AesCoder achieves state-of-the-art aesthetic quality, which necessitates a trade-off in general algorithmic performance, (2) the model **retains functional coding competence** without suffering from complete catastrophic forgetting, and (3) the observed regressions are consistent with the behavior of models undergoing intensive fine-tuning for highly specific domains.

## I Failure Case Study

### I.1 Non-executable & Visually Incorrect

> User Prompt: Design a website about 1+1.

The model's output is as follows:

```
2
```

**Analysis.** This output is **non-executable and visually incorrect**. The user requested a webpage that displays "1+1," but the model instead produced the direct result "1+1=2," and the output is not even in a valid HTML format. Therefore, **the Execution Agent fails this sample, and neither the Static Aesthetic Agent nor the Interactive Aesthetic Agent is triggered**.

**Final reward.** $r_{exec} = -1$, no $r_{static}$ and $r_{interact}$ (not triggered). The final reward is $r = -1$.

### I.2 Non-executable & Visually Correct

> User Prompt: Make a simple personal website.

The HTML code of the webpage is as follows (partial snippet):

```html
<!DOCTYPE html>
<html lang="en">
<head>

...

        <div class="section" id="contact-section">
            <h2>Contact</h2>
            <p>Feel free to reach out to me!</p>
            <div class="contact-links">
                <a href="mailto:xxxx@example.com">Email</a>
                <a href="https://github.com/">GitHub</a>
                <a href="https://linkedin.com">LinkedIn</a>
            </div>
        </div>

        <div class="section" id="contact-section">
            <h2>Follow Me</h2>
            <p>Connect with me on social media!</p>
        </div>
    </div>
</body>
</html>
```

The webpage screenshot is Figure 7.

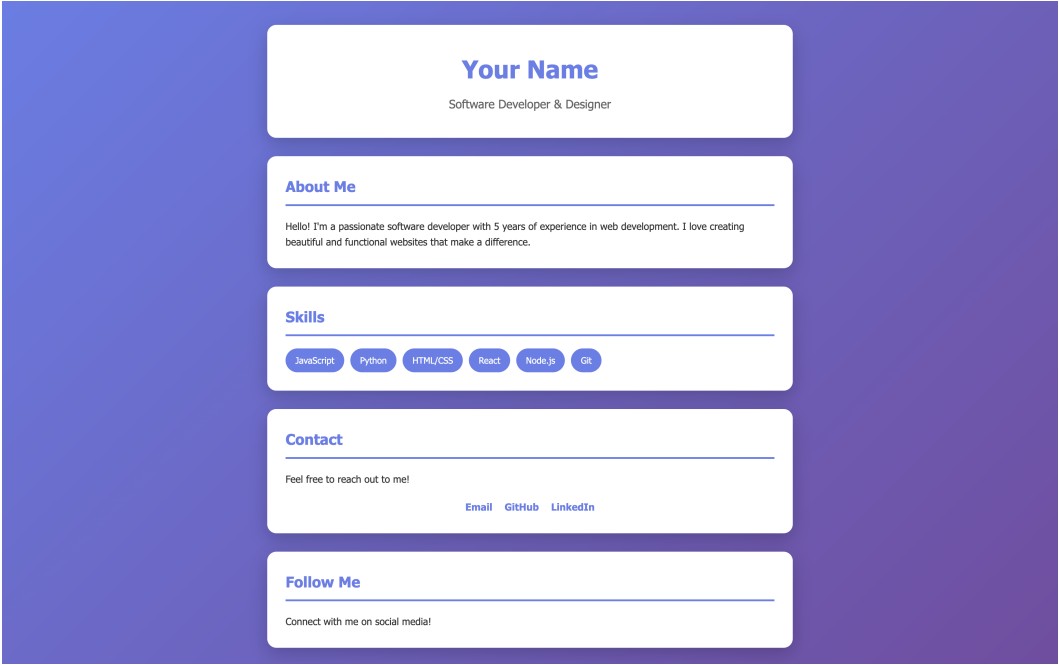

Figure 7: A webpage screenshot of non-executable but visually correct code.

**Analysis.** A close inspection of the page's HTML source code (partial snippet shown below) reveals that both `<div>` elements use the same `id="contact-section"` (highlighted in red). In standard HTML, IDs must be unique within a document, and this violation is correctly detected by HTMLHint during executability validation. **Although the rendered webpage appears visually normal—with no immediately noticeable defects from a screenshot perspective—the underlying HTML is structurally incorrect.**

Because of this, the Execution Agent fails the sample and assigns a reward of $r_{\text{exec}} = -1$. As a result, neither the Static Aesthetics Agent nor the Interactive Aesthetics Agent is triggered, ensuring that the model receives no credit for visually plausible but syntactically invalid outputs. This strict gating mechanism enforces the production of standards-compliant, well-structured HTML, preventing the model from exploiting aesthetics to bypass fundamental correctness.

**Final reward.** $r_{exec} = -1$, no $r_{static}$ and $r_{interact}$ (not triggered). The final reward is $r = -1$.

## I.3 EXECUTABLE BUT VISUALLY BROKEN

> User Prompt: Make a simple webpage of an Italian restaurant.

The webpage screenshot is Figure 8

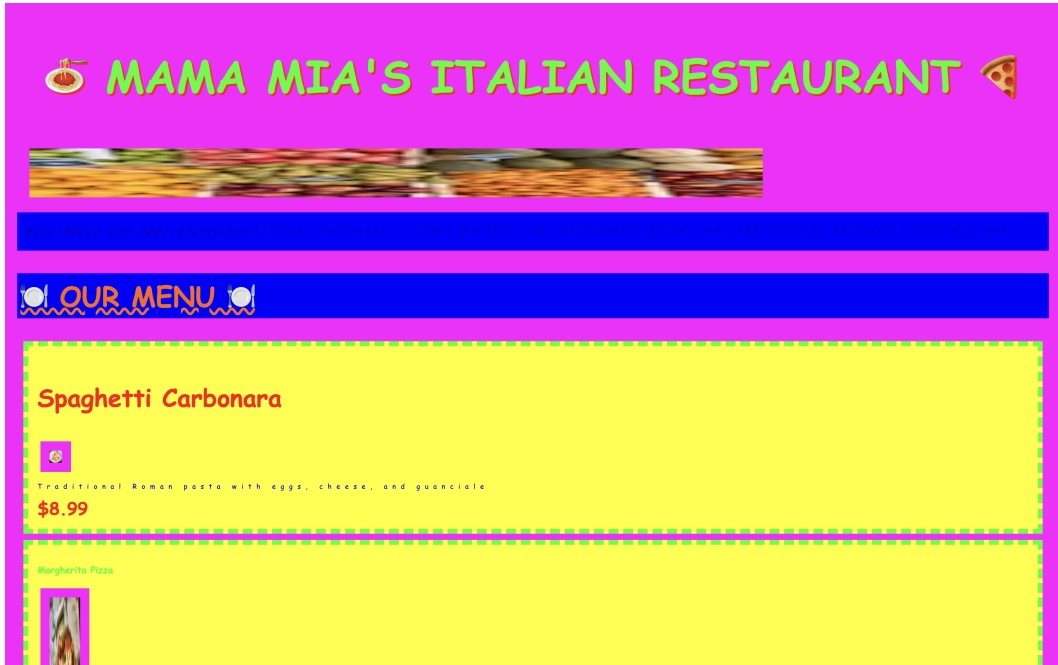

Figure 8: A webpage screenshot of executable but visually broken code.

**Analysis.** The model generates HTML code that is fully syntactically valid and passes executability checks. However, the rendered webpage reveals **severely degraded visual quality**. As shown in the screenshot, the design suffers from clashing color combinations (purple–red, blue, and yellow), low-resolution imagery, and text elements with fonts that are far too small for comfortable reading.

Because of these issues, **the Static Aesthetics Agent assigns a very low score (18 points)**, reflecting that although the page minimally aligns with the user's instruction, its overall visual presentation is substandard. Furthermore, since the webpage contains no interactive elements, **the Interactive Aesthetics Agent correctly assigns a score of 0**.

This example demonstrates that even when the model produces syntactically correct HTML, the reward system does not overlook poor design quality: executability alone is insufficient, and the Static and Interactive agents provide crucial complementary supervision.

**Final reward.** $r_{exec} = 1$, $r_{static} = -0.64$ and $r_{interact} = -1$. The final reward is $r = -0.512$.

## I.4 VISUALLY GOOD BUT FUNCTIONALLY WRONG

> User Prompt: Design a website for me to manage my daily spending records.

The webpage screenshot is Figure 9

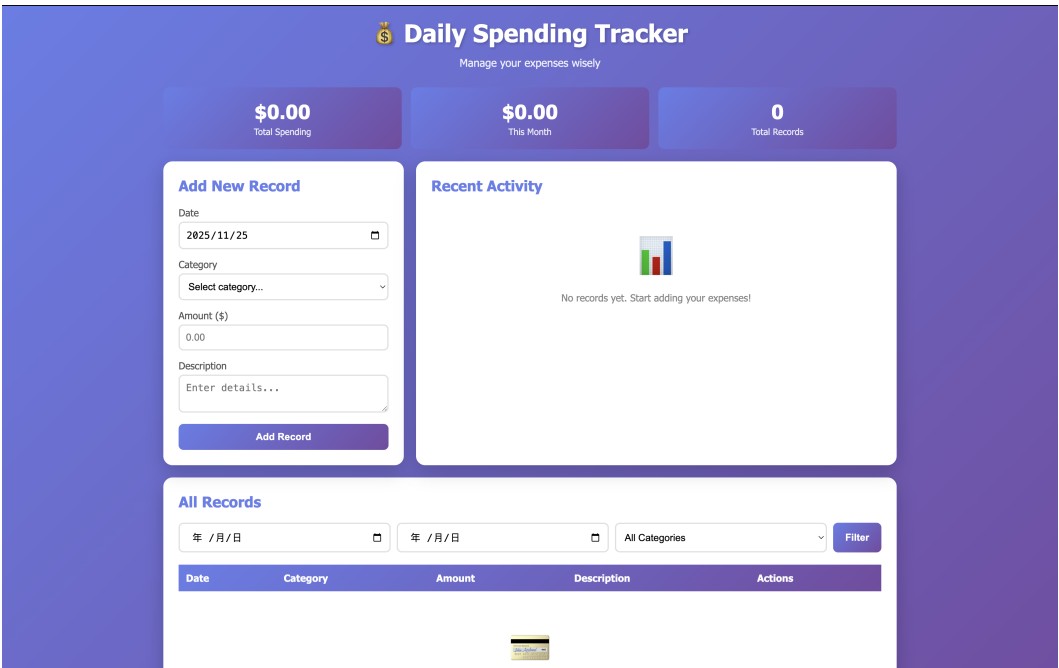

Figure 9: A webpage screenshot of visually good but functionally wrong code.

**Analysis.** The model generates HTML code that is **both syntactically valid and visually well-designed**. The rendered webpage appears polished, with a clean layout and aesthetically pleasing styling. As a result, **the Static Aesthetics Agent assigns a high score of 81 points**.

However, despite its strong visual presentation, the webpage **fails to satisfy the functional requirements** specified by the user. The user requested a webpage for recording daily expenses, yet the "Add Record" button is entirely non-functional: clicking it does not log the expense or trigger any observable state change. This constitutes a clear violation of the expected interactive behavior. On the other hand, **certain elements—such as the date-selection component—do respond correctly**. Consequently, the Interactive Aesthetics Agent assigns a partial score of 1, **reflecting the presence of some functioning elements but penalizing the non-operational core feature**.

This case illustrates why interactive evaluation is essential: a webpage may look correct and receive a high static score, but without functional alignment, the overall reward must be significantly reduced.

**Final reward.** $r_{exec} = 1$, $r_{static} = 0.62$ and $r_{interact} = 0$. The final reward is $r = 0.596$.

