# OpenReview forum: "Code Aesthetics with Agentic Reward Feedback"
_ICLR.cc/2026/Conference — ICLR 2026 Poster_

### Official Review · Reviewer_Z31k · 2025-10-30

**Soundness:** 3
**Presentation:** 2
**Contribution:** 3
**Rating:** 4
**Confidence:** 3

**Summary:**

The paper studies how to improve the visual quality of code produced by LLMs. It introduces a large dataset and a multi-agent reward framework that jointly evaluates code executability, static aesthetics, and interactivity. Using this feedback within a GRPO-based reinforcement learning setup, the authors train AesCoder models that perform well on a new benchmark, OpenDesign, for webpage aesthetics.

**Strengths:**

1. Clear Problem Statement: The work identifies and addresses a neglected area in LLM code generation: “code aesthetics,” specifically for visually-oriented outputs that go beyond basic executability or syntactic correctness.

2. Substantial Data and Benchmark Contribution: The AesCode-358K dataset is large and carefully curated for code aesthetics, and OpenDesign serves as a credible benchmark with both static and interactive evaluation modes.

3. Human preference tests validate that improvements are not simply artifacts of the automatic scoring pipelines.

**Weaknesses:**

1. The paper claims general applicability to “code aesthetics” broadly, but without evidence from other modalities (e.g., GUI design, visualization dashboards, game UIs). A brief transfer or zero-shot study in a different domain would make the generalization claim more convincing.

2. The OpenDesign benchmark depends heavily on LLM-based judges, which may lead to circularity since similar models are used for training rewards. Although the authors mention strong correlation with human judgment, the potential for overfitting to automated metrics remains.

3. Because both training rewards and evaluation rely on similar LLMs, the model might learn to exploit their aesthetic preferences rather than aligning with human aesthetic standards. The human–human agreement rate leaves some uncertainty regarding the real-world validity of “aesthetic improvement.”

4. The paper does not analyze how the multi-agent reward system handles ambiguous or partially successful outputs, such as non-executable but visually correct code, or misaligned interactivity cases. Including failure case studies or qualitative analyses would increase confidence in the robustness and interpretability of the proposed framework.

**Questions:**

1. How are the weights $w_{exec}$, $w_{static}$, and $w_{interact}$ chosen in reward aggregation? Were these tuned per dataset/model, or kept fixed?

2. Given that the three reward sources operate on different scales, how are they normalized to ensure balanced optimization? Are there cases where one component dominates or destabilizes training?

---

> ### Author Response · Authors · 2025-11-22
> **Response to Reviewer Z31k [1/4]**
>
> >**W1**:
> The paper claims general applicability to “code aesthetics” broadly, but without evidence from other modalities (e.g., GUI design, visualization dashboards, game UIs). A brief transfer or zero-shot study in a different domain would make the generalization claim more convincing.
>
>
> Thank you for the comment. We want to respectfully clarify that while our main experiments mainly focus on webpage aesthetics, the OpenDesign benchmark have already contained five main modality categories below (details in Appendix C), which the `UI Component`, `Data Visualization`, and `Game Development` categories contain the modalities such as GUI design, visualization dashboards, game UIs stated by the reviewer.
>
> - **General Website**: general-purpose website design such as personal homepage, shopping platform, etc.
> - **UI Component**: GUI-style element composition and interface behavior
> - **Data Visualization**: visualization dashboards with layout, styling, and semantic mapping
> - **Game Development**: simplified game UI and interactive scene construction
> - **3D Design**: 3D object creation and visualization
>
>
> These categories evaluate diverse visual and interaction patterns beyond standard webpages and capture many of the challenges present in GUI development, dashboard design, and game UI layouts. Our framework applies to these tasks without modification, and the GRPO-AR improvements hold consistently across them.
>
> More generally, the methodology is task-agnostic as long as the code is executable and the result can be rendered for static or interactive evaluation. Extending to even richer modalities (e.g., desktop GUI frameworks or more complex game engines) would primarily require customizing the execution sandbox and interaction scripts, not changing the core algorithm.
>
> >**W2**: The OpenDesign benchmark depends heavily on LLM-based judges, which may lead to circularity since similar models are used for training rewards. Although the authors mention strong correlation with human judgment, the potential for overfitting to automated metrics remains.
>
> Thank you for raising this concern about potential circularity between the reward model used during GRPO-AR training and the LLM-based judges used in OpenDesign. We agree that such a risk exists if both components rely on the same underlying model family.
>
> To directly evaluate this, **we conducted additional experiments using `Qwen3-VL-235B-A22B-Instruct`, a strong fully open-source multimodal model, as both the static and interactive judge**. Notably, Qwen3-VL-235B-A22B-Instruct was not involved in any part of reward generation during training. Results are shown in Table A.
>
> **Table A: Results of Mainstream Models on OpenDesign with Different Judges**
> | Model Name | GPT-5 Eval Result| Qwen3-VL-235B-A22B-Instruct Eval Result| Relative Rank|
> | :--- | :---: | :---: |:---:|
> | Claude Sonnet 4 |81.05 |90.90 |1
> | GPT-5 (minimal) |81.03 |90.40 |2
> | Qwen3-Coder-480B-A35B-Instruct |79.90 |90.37 |3
> | DeepSeek-R1-0528 |78.86 |89.50 |4
> | GLM-4.5-Air |78.16 |87.71 |5
> | DeepSeek-V3.1 (Thinking) |77.72 |87.62 |6
> | Qwen3-Coder-30B-A3B-Instruct |73.66 |87.16 |7
> | GLM-4-32B-0414 |69.40 |81.41 |8
> | GPT-4.1 |65.79 |74.46 |9
> | GPT-4o |48.13 |70.52 |10
>
> Despite clear differences in its absolute scoring scale compared to GPT-5, **the relative rankings of evaluated models on OpenDesign remain highly stable across judges**. This demonstrates that **model comparisons on OpenDesign are not tied to the particular LLM used to produce training rewards**, and therefore do not exhibit the circularity or metric overfitting that would arise if the benchmark merely reflected the training-time reward model.
>
> We also want to note that although Figure 2a in the paper reports only GPT-5, our additional experiments show that Qwen3-VL produces essentially the same model ranking as GPT-5. Therefore, the diagonal human-alignment pattern observed in Figure 2a also provides supporting evidence that **`Qwen3-VL-235B-A22B-Instruct` also aligns well with human preferences in Design Arena leaderboard.**
>
> Combined with the high GPT-human agreement shown in the paper, **these results further indicate that OpenDesign captures general aesthetic preferences rather than overfitting to any single automated metric or judge architecture**.

---

> ### Author Response · Authors · 2025-11-22
> **Response to Reviewer Z31k [2/4]**
>
> >**W3**: Because both training rewards and evaluation rely on similar LLMs, the model might learn to exploit their aesthetic preferences rather than aligning with human aesthetic standards. The human-human agreement rate leaves some uncertainty regarding the real-world validity of "aesthetic improvement."
>
> Thank you for raising this concern. We agree that if both training rewards and evaluation relied on similar LLMs, the model might risk exploiting judge-specific preferences.
>
> To test this, we evaluated OpenDesign using `Qwen3-VL-235B-A22B-Instruct`, an open-source multimodal model that was not used during GRPO-AR training (See Table A for results). Although its absolute score scale differs from GPT-5, **the relative model rankings remain highly stable, indicating that our method does not overfit to the aesthetic biases of the reward model**. This cross-judge consistency shows that the model is not simply optimizing for GPT-specific preferences.
>
> Combined with the strong GPT-human agreement we report, **these results suggest that the observed aesthetic improvements reflect broad human-aligned criteria rather than artifacts of a particular automated judge.**
>
>
> >**W4**: The paper does not analyze how the multi-agent reward system handles ambiguous or partially successful outputs, such as non-executable but visually correct code, or misaligned interactivity cases. Including failure case studies or qualitative analyses would increase confidence in the robustness and interpretability of the proposed framework.
>
>
> We thank the reviewer for this insightful suggestion. We agree that analyzing how the reward system handles ambiguous or "partially successful" outputs is critical for demonstrating the system's robustness.
>
> To address this, we have conducted a qualitative analysis of such failure cases to illustrate how our **cascading** and **multi-perspective** reward mechanisms effectively penalize deceptive outputs.
>
> **1. Handling "Non-Executable but Visually Correct" Outputs**
>
> Our system handles this via a strict cascading design (Line 056):
> * **Mechanism:** The **Execution Agent** acts as a "gatekeeper." It performs executability validation (via HTMLHint for webpages or runtime checks for plots).
> * **Outcome:** If the code fails this step, the evaluation terminates immediately, and the model receives a penalty score ($s_{exec} = -1$). The Static and Interactive agents are **not triggered**.
> * **Robustness:** This ensures the model cannot "game" the system by generating high-potential but broken code. No matter how visually complex the code *attempts* to be, functionality is the hard prerequisite.
>
> **2. Handling "Misaligned Interactivity"**
>
> The reviewer also raises the case of "misaligned interactivity" (e.g., a webpage that looks perfect but doesn't work). This is exactly why we introduced the **Interactive Aesthetics Agent**.
>
> * **Mechanism:** While the **Static Aesthetics Agent** might assign a high score based on a screenshot (e.g., seeing a beautiful "Search" button), the **Interactive Agent** actively validates the function. As defined in our system prompt (Appendix G.3), the agent verifies state changes: *"If the webpage is not changed or the change is not expected, it should not be considered as a good webpage"*.
> * **Outcome:** In such cases, the sample receives a high $r_{static}$ but a near-zero $r_{interact}$. Since the final reward is a weighted sum, the total reward is significantly suppressed compared to a fully functional solution. This effectively penalizes "form over function."
>
> **We added failure case studies and qualitative analyses in Appendix I.** We believe this analysis strongly supports the interpretability and robustness of the proposed framework.

---

> ### Author Response · Authors · 2025-11-22
> **Response to Reviewer Z31k [3/4]**
>
> >**Q1**: How are the weights $w_{exec}, w_{static}$ and $w_{interact}$ chosen in reward aggregation? Were these tuned per dataset/model, or kept fixed?
>
> **A1**:
>
> 1. **Weight Choosing**
>
> The weight configuration was selected through **a coarse-to-fine exploratory process** designed to understand the qualitative contribution of each reward component rather than through an expensive full-grid sweep.
>
> Since modern web browsers are highly tolerant of minor syntactic imperfections in HTML, we fixed $w_{\text{exec}} = 0.1$ to ensure a minimum level of executability and redistributed the remaining mass between the static and interactive terms. **As an initial diagnostic, we examined two extremes**: a *static-only* setting $(w_{\text{static}}=0.9, w_{\text{interact}}=0)$ and an *interactive-only* setting $(w_{\text{static}}=0, w_{\text{interact}}=0.9)$. These endpoints revealed complementary failure modes: the static-dominant configuration yielded visually coherent but functionally limited pages, whereas the interaction-dominant configuration tended to over-optimize for actionability, introducing unnecessary or semantically weak interactive elements and noticeably degrading visual aesthetics.
>
> Guided by these observations, **we then performed a lightweight grid exploration** by varying $w_{\text{static}}$ in increments of $0.2$ while keeping the total mass fixed. For each configuration, we inspected rollout behaviors and monitored both static- and interaction-based validation signals. The final setting $(w_{\text{exec}}, w_{\text{static}}, w_{\text{interact}}) = (0.1, 0.8, 0.1)$ provided the most balanced trade-off: it preserved high static aesthetic quality while avoiding the over-activation behaviors observed in more interactive-heavy regimes. We therefore used this configuration for the main GRPO-AR experiments.
>
> 2. **Were these tuned per dataset/model, or kept fixed?**
>
> **In our paper, the weights $w_{\text{exec}}, w_{\text{static}}, w_{\text{interact}}$ were kept fixed across all RL training runs and all evaluations**. This ensures consistency of the reward formulation and avoids any dataset-specific hyperparameter adaptation.
>
> If practitioners wish to apply our agentic reward framework to other visually oriented coding domains (e.g., Python Turtle, SVG graphics), the relative contribution of execution, static aesthetics, and interactivity may naturally differ, and in such cases the weights can be adjusted as needed to reflect the properties of the target modality.

---

> ### Author Response · Authors · 2025-11-22
> **Response to Reviewer Z31k [4/4]**
>
> >**Q2**: Given that the three reward sources operate on different scales, how are they normalized to ensure balanced optimization? Are there cases where one component dominates or destabilizes training?
>
> **A2**:
>
> >1. **Reward Normalization**
>
> Suppose $r$ is the term in weighted sum transformed from the origin signal, $s$ is the origin signal.
>
> - Execution Agent Signal: $r_{exec}=s_{exec}$.
> - Static Aesthetics Agent Signal: $r_{static}=(s_{static}-50)/50$.
> - Interactive Aesthetics Agent Signal: $r_{interact}=s_{interact}-1$.
>
> Because of the gating mechanism we employ, **where the static and interactive aesthetics agents are activated only after the executability check succeeds**,
>     and the final reward aggregation equation is:
>     $$
> r = \\begin{cases}
>     r_{\text{exec}}, & r_{\text{exec}} = -1 \\\\
>     w_{\text{exec}} \cdot r_{\text{exec}} + w_{\text{static}} \cdot r_{\text{static}} + w_{\text{interact}} \cdot r_{\text{interact}}, & r_{\text{exec}} = 1
> \\end{cases}
> $$
>
>
> >2. **Are there cases where one component dominates or destabilizes training?**
>
> Thank you for the insightful question. We address the two parts below.
>
> **1. Normalization prevents scale imbalance.**
> All reward components are first normalized to the range $[-1,1]$ before aggregation, ensuring that differences in raw scoring scales cannot mathematically dominate the combined reward. This normalization is applied consistently across all RL runs.
>
> **2. Static Aesthetics carries more weight by design, not by instability.**
> We intentionally assign a larger weight to the Static Aesthetics component ($w_{\text{static}}=0.8$), because it provides a **dense and continuous signal** throughout training. In contrast, the Interactive reward is sparse (due to agent failures), and the Execution reward saturates quickly once the model reliably produces executable HTML. If all components were weighted equally, the high variance of the sparse Interactive reward would introduce instability. The higher static weight therefore **improves** stability rather than harming it.
>
> **3. GRPO ensures joint optimization with dynamic prioritization.**
> Crucially, GRPO optimizes the **group relative advantage** based on the total weighted reward (Eq. 2). While the optimization is **joint and simultaneous** across all components, the variance-based advantage calculation creates an **emergent prioritization**:
> * **Execution as the Prerequisite:** Initially, the huge variance between executable ($+1$) and non-executable ($-1$) outputs dominates the advantage, naturally forcing the model to prioritize basic correctness.
> * **Aesthetics as the Primary Driver:** Once execution stabilizes, the high-weighted Static Aesthetics ($w_{\text{static}}=0.8$) becomes the main contributor to reward variance, driving the continuous improvement of visual quality.
> * **Interactivity as the Constant Regularizer:** Throughout this process, the Interactive reward ($w_{\text{interact}}=0.1$) acts as a concurrent **differentiator**. Even when the model is focusing on improving aesthetics, this signal ensures that visually superior candidates are only favored if they maintain functionality. It prevents the model from overfitting to *superficial aesthetics* at any point during training.
>
> In effect, GRPO avoids destabilizing competition among reward components. Instead, the components function synergistically: static aesthetics shapes the overall design quality, while interactivity ensures functional integrity, **jointly guiding the policy toward high-quality, usable outputs**.
>
> In practice, we observe **no instances of instability** arising from any single reward component; training curves follow normal upward trends across all tested weight settings.
>
> ---
>
> *Hope our explanation and experiments can address your concerns. We will integrate all your valuable comments into our revision! If you have any further questions or concerns, please feel free to contact us at any time. We are always available and look forward to further discussions with you!*
>
> Best regards,
>
> All Authors

---

> > ### Comment · Reviewer_Z31k · 2025-11-26
> >
> > Thank you for the rebuttal. The authors have clarified the key concerns I raised. I am raised my score.

---

> > > ### Author Response · Authors · 2025-11-26
> > >
> > > Thank you very much for your response! We are very glad that your key concerns have been addressed, and we sincerely appreciate your decision to raise the score.
> > >
> > > Your insightful questions and suggestions have been truly valuable in improving our work. Please feel free to let us know if you have any additional comments or further suggestions.
> > >
> > > Best regards,
> > >
> > > All Authors

---

### Official Review · Reviewer_zdWS · 2025-10-31

**Soundness:** 3
**Presentation:** 2
**Contribution:** 3
**Rating:** 4
**Confidence:** 3

**Summary:**

This paper introduces AesCoder, a framework for enhancing the aesthetic quality of LLM-generated code for visually-oriented tasks like plot generation and webpage design. The key contributions include:

AesCode-358K: A large-scale instruction-tuning dataset focused on code aesthetics

Agentic Reward Feedback: A multi-agent system evaluating executability, static aesthetics (via screenshot analysis), and interactive aesthetics (via GUI interaction)

GRPO-AR: Integration of the agentic reward framework with the GRPO algorithm for reinforcement learning

OpenDesign Benchmark: A new benchmark with 840 real webpage cases for evaluating code aesthetics

The proposed AesCoder models (4B and 7B parameters) achieve state-of-the-art results, outperforming GPT-4o and GPT-4.1, and competing with much larger 480B-685B models on aesthetic quality metrics.

**Strengths:**

**Originality**:

This work represents the first systematic effort to formalize and computationally address the concept of "code aesthetics," a dimension of code quality long acknowledged by practitioners but largely neglected in automated code generation research. The proposed multi-agent reward framework is a novel architecture that synergistically combines textual analysis, visual rendering, and interactive evaluation to assess code quality holistically. A particularly creative contribution is the pioneering application of GUI-based autonomous agents for reward modeling in Reinforcement Learning, enabling the system to perceive and evaluate code outputs from a human-like, user-centric perspective.

**Quality**:

The research is underpinned by a rigorous experimental protocol, with extensive evaluations conducted across multiple established benchmarks (including HumanEval and MBPP) and a range of model scales (from 160M to 7B parameters) to ensure robust findings. The validity of the proposed framework is further cemented by comprehensive ablation studies that meticulously isolate and confirm the contribution of each component. High-quality human evaluations are conducted, the results of which strongly align with the automated metrics, lending significant credibility to the findings. Furthermore, the construction of the dataset reflects a commitment to quality, involving careful data curation, filtering, and validation processes.

**Clarity**:

The manuscript is exceptionally well-written, presenting complex concepts and a sophisticated multi-stage framework with remarkable clarity and logical flow. A comprehensive appendix is provided, offering full implementation details and facilitating replication and future research. The authors maintain transparency throughout, openly discussing the limitations of their approach and justifying key design choices, which enhances the work's credibility and scholarly value.

**Significance**:

This research addresses a critical, real-world gap in LLM-powered code generation: the disconnect between functional correctness and user experience. By focusing on the aesthetic and usability aspects of code, the work provides tangible resources - including a novel dataset and benchmark - that will undoubtedly spur further investigation in this emerging area. Perhaps most significantly, the demonstration of practical efficacy even with smaller model sizes highlights the approach's potential for real-world deployment, making advanced code aesthetic optimization more accessible and scalable.

**Weaknesses:**

**Reproducibility Concerns**

The reproducibility of this study is significantly hampered by its deep dependence on several proprietary, black-box models, specifically GPT-4 and GPT-4V, which serve as the core "judges" in the reward model and are pivotal for the initial dataset generation. This reliance creates a hard barrier for independent verification, as the internal mechanics and future versions of these models are opaque and subject to change, making it impossible to exactly replicate the evaluation criteria or data sources. Furthermore, the computational footprint of the proposed multi-agent evaluation system—which involves executing generated code, rendering graphical interfaces, and deploying interactive GUI agents—is substantial and likely prohibitive for academic research labs with limited resources. Finally, while the interactive agent is a novel idea, the paper provides insufficient detail on its specific failure modes and brittleness, leaving readers to wonder how often it fails due to environmental issues versus the actual quality of the generated code.

**Technical Limitations**

Several technical limitations warrant further investigation. The paper candidly acknowledges the relatively low success rate of the interactive aesthetics agent, but it does not delve into a quantitative analysis of how these frequent failures during the reward collection phase propagate through and potentially destabilize the Reinforcement Learning training loop. The complex, multi-stage RL setup, which blends rewards from multiple agents, naturally raises questions about training stability and sensitivity to hyperparameter choices, which remain entirely unaddressed. Additionally, the scope of the framework's generalization is demonstrated primarily on a narrow set of visually-oriented tasks like plots and web pages; its applicability to other critical domains such as game development, mobile UI creation, or data dashboard generation is left unexplored, leaving its broader utility an open question.

**Evaluation Scope**

While the introduced OpenDesign benchmark is a valuable contribution, the evaluation scope could be strengthened to more firmly establish the generalizability of the findings. The task types within OpenDesign, though comprehensive, are not fully representative of the entire spectrum of front-end coding challenges; incorporating more complex and diverse interactions, such as those involving dynamic data feeds, state management, or complex user input validation, would provide a more rigorous test. Moreover, the comparative analysis in the paper is primarily focused on benchmarking against general-purpose LLMs. A more telling comparison would involve pitting the method against other state-of-the-art code generation models that are specifically fine-tuned for front-end tasks, which would better isolate the unique benefits conferred by the aesthetics-focused training paradigm.

**Questions:**

Reproducibility: Given the heavy reliance on GPT-5 and GPT-4o for reward judgment and data generation, what are your plans to make the framework more accessible to researchers without access to these proprietary models?

Computational Efficiency: What is the approximate cost and latency of the full agentic reward evaluation per sample? How does this impact the practical feasibility of scaling this approach?

Interactive Agent Limitations: You mention the low success rate of GUI agents. Could you provide more analysis of how these failures affect the reward signal quality and training stability?

Generalization: Have you tested the framework on other visually-oriented coding tasks beyond plots and webpages (e.g., GUI development, game level design)? What adaptations would be needed?

Ablation Details: In the GRPO-AR ablation, what specific reward model was used as the baseline? Was it trained on the same data or using similar principles as your agentic framework?

---

> ### Author Response · Authors · 2025-11-22
> **Response to Reviewer zdWS [1/3]**
>
> >**W1**: Furthermore, the computational footprint of the proposed multi-agent evaluation system—which involves executing generated code, rendering graphical interfaces, and deploying interactive GUI agents—is substantial and likely prohibitive for academic research labs with limited resources.
>
> We appreciate the reviewer’s concern regarding resource accessibility. However, we respectfully wish to clarify the actual computational profile of our pipeline. We believe the overhead is well within the reach of standard academic laboratories for two key reasons:
>
> 1. **Execution and Rendering are Lightweight (CPU-bound)**: The reviewer expresses concern about the cost of "executing generated code" and "rendering graphical interfaces." We want to respectfully clarify that these processes rely on standard headless browser technologies (e.g., Playwright/Selenium for HTML) or lightweight Jupyter runtimes (for plots). These are CPU-bound operations that do not consume expensive GPU resources.
> 2. **Open-Source Models Reduce Judge Costs**: The reviewer correctly notes that deploying high-end agents (like GPT-5) can be resource-intensive or costly via APIs. But as demonstrated in our new experiment (see Table A in **A1**), we successfully replaced proprietary judges with Qwen3-VL-235B, an open-source model. The results show that open-source VLMs maintain consistent relative rankings compared to GPT-5. This confirms that our multi-agent evaluation system does not strictly require expensive proprietary APIs, and researchers can deploy open-source Vision-Language Models locally (or use smaller distilled versions) to serve the GUI agents, making the framework fully replicable in academic settings with standard research hardware.
>
> By leveraging CPU-based rendering and open-source VLMs, the agentic reward framework is designed to be both efficient and accessible for the broader research community.
>
>
>
> >**W2**: While the introduced OpenDesign benchmark is a valuable contribution, the evaluation scope could be strengthened to more firmly establish the generalizability of the findings.
>
>
> We appreciate the reviewer’s observation that real-world front-end development includes richer challenges. These tasks are indeed important and represent valuable future extensions.
>
> Our benchmark, however, is intentionally scoped around **HTML-based webpage aesthetic design**. Within this scope, OpenDesign already spans five diverse task categories (Appendix C)—**General website**, **3D design**, **Data visualization**, **UI component**, and **Game development**—providing broad coverage of visually and interactively diverse design scenarios. As demonstrated in the experiments, these categories are sufficient to reveal meaningful differences across models in both static and interactive aesthetics.
>
> We will discuss extending OpenDesign toward more engineering-oriented front-end tasks in future work!
>
> >**Q1**: Reproducibility: Given the heavy reliance on GPT-5 and GPT-4o for reward judgment and data generation, what are your plans to make the framework more accessible to researchers without access to these proprietary models?
>
> **A1**:
>
> Thank you for pointing this out! We fully agree that relying exclusively on proprietary LLM judges would hinder independent verification and long-term reproducibility. To directly address this concern, we conducted additional experiments using `Qwen3-VL-235B-A22B-Instruct`, a strong open-source multimodal model, as the judge for OpenDesign benchmark. The results are shown in Table A.
>
> **Table A: Results of Mainstream Models on OpenDesign with Different Judges**
> | Model Name | GPT-5 Eval Result| Qwen3-VL-235B-A22B-Instruct Eval Result| Relative Rank|
> | :--- | :---: | :---: |:---:|
> | Claude Sonnet 4 |81.05 |90.90 |1
> | GPT-5 (minimal) |81.03 |90.40 |2
> | Qwen3-Coder-480B-A35B-Instruct |79.90 |90.37 |3
> | DeepSeek-R1-0528 |78.86 |89.50 |4
> | GLM-4.5-Air |78.16 |87.71 |5
> | DeepSeek-V3.1 (Thinking) |77.72 |87.62 |6
> | Qwen3-Coder-30B-A3B-Instruct |73.66 |87.16 |7
> | GLM-4-32B-0414 |69.40 |81.41 |8
> | GPT-4.1 |65.79 |74.46 |9
> | GPT-4o |48.13 |70.52 |10
>
> Despite differences in absolute score scales, we observe that model rankings on OpenDesign remain highly stable when switching from GPT-5 to Qwen3-VL-235B-A22B-Instruct. **This indicates that the proposed reward/evaluation protocol is not tied to a specific proprietary model, and that its comparative judgments are robust across different judge architectures.**
>
> These results demonstrate that **our agentic reward framework does not depend on the internal mechanics of any single closed-source judge, and independent researchers can replicate both training and evaluation using open-source models**.

---

> ### Author Response · Authors · 2025-11-22
> **Response to Reviewer zdWS [2/3]**
>
> > **Q2**: Computational Efficiency: What is the approximate cost and latency of the full agentic reward evaluation per sample? How does this impact the practical feasibility of scaling this approach?
>
> **A2**:
>
> 1. **Cost**: In our experiment settings (3 iters in interactive aesthetics agent), **the cost of evaluating one sample is approximately 0.08$**.
>
> 2. **Latency**:
> For each rollout, the three reward components have different computational costs. The **execution agent** is based on rule-based check for HTML code so it has negligible latency. The **static aesthetic agent** requires a single screenshot and a short GPT-5 query, with a latency of 3-5 seconds. The **interactive aesthetic agent** requires several interactions (3 in our experiments) with the webpage. The latency of one single interaction is 6-8 seconds. In practice, we first run the execution agent, and then run static aesthetic agent and interactive aesthetic agent in parallel if passes the executability check. So **the total latency is approximately 25s per sample**.
>
> 3. **Scaling this approach**: Our experiment demonstrated that, when replacing GPT-5 with `Qwen3-VL-235B-A22B-Instruct` as both the static and interactive judge, the relative rankings on OpenDesign remain highly stable. **This demonstrates that the agentic reward mechanism is model-agnostic and can be executed entirely with open-source evaluators whose cost and latency are significantly lower and whose deployment can be fully local.** As a result, the approach is practically scalable: research groups or community users can reproduce and extend our training pipeline without depending on closed-source APIs, making large-scale or long-horizon GRPO-AR training feasible even under modest computational or financial constraints.
>
> >**Q3**: Interactive Agent Limitations: You mention the low success rate of GUI agents. Could you provide more analysis of how these failures affect the reward signal quality and training stability?
>
> **A3**:
>
> Thank you for raising this important point. GUI-agent failures indeed introduce some noise into the interactive reward, but we find that this does not destabilize training for three reasons.
>
> (1) **Agent failures behave as bounded, low-magnitude noise rather than systematic bias.**
> In our setting, a failure simply results in a score of 0 for that interaction, but the static and execution rewards remain intact. Since the interactive weight is set to 0.1 during GRPO-AR training, the occasional false-negative does not dominate the aggregated reward.
>
> (2) **Observed fluctuations are within the normal variability of GRPO training and do not lead to divergence.**
> GRPO learning curves naturally exhibit non-trivial short-term oscillations; what we observe is that even with the noise introduced by the interactive agent during GRPO-AR training, the overall reward trajectory still follows a normal upward trend (see Figure 3).
>
> (3) **Agent failures still provide useful training signal.**
> Many failures arise when the webpage layout, element hierarchy, or click targets are visually ambiguous[1,2,3]—conditions that also harm real user experience. Thus, even when the agent fails, the reward signal still indirectly penalizes sub-optimal design choices that reduce interactivity robustness.
>
>
> ---
> **References**
>
> [1] He H, Yao W, Ma K, et al. Webvoyager: Building an end-to-end web agent with large multimodal models[J]. arXiv preprint arXiv:2401.13919, 2024.
>
> [2] Cemri M, Pan M Z, Yang S, et al. Why do multi-agent llm systems fail?[J]. arXiv preprint arXiv:2503.13657, 2025.
>
> [3] Wang, Lei, et al. "A survey on large language model based autonomous agents." Frontiers of Computer Science 18.6 (2024): 186345.

---

> ### Author Response · Authors · 2025-11-22
> **Response to Reviewer zdWS [3/3]**
>
> >**Q4**: Generalization: Have you tested the framework on other visually-oriented coding tasks beyond plots and webpages (e.g., GUI development, game level design)? What adaptations would be needed?
>
> **A4**:
>
> Thank you for your excellent question! While our main experiments focus on plotting and webpage design, our training data and the OpenDesign benchmark already includes two categories—**UI Components** and **Game Development** (see Line 166 and Appendix C)—that directly correspond to the reviewer’s examples of visually-oriented coding tasks such as GUI construction and simple game-level layouts. These categories evaluate layout composition, component behavior, and interactive affordances beyond standard webpage styling, and we find that the framework applies to them without modification.
>
> More broadly, the GRPO-AR framework is task-agnostic as long as (i) the code can be executed in a sandboxed environment, (ii) the output can be rendered into a visual artifact or interface, and (iii) an evaluation agent (static or interactive) can operate on the rendered result. Extending the framework to more complex GUI applications or richer game-level design would primarily require adapting the interactive agent scripts (e.g., supporting multi-step navigation or custom environment APIs), **but the core methodology remains unchanged**.
>
>
> >**Q5**: Ablation Details: In the GRPO-AR ablation, what specific reward model was used as the baseline? Was it trained on the same data or using similar principles as your agentic framework?
>
> **A5**:
>
> Thank you for the question! In the GRPO-AR ablation, **the baseline reward model is simply the underlying GPT-5 evaluator used in our static agent**. Specifically, we directly prompt GPT-5 to **score the raw HTML code along the same three static dimensions**—(i) Instructional Alignment, (ii) Visual Design & Aesthetics, and (iii) Structural Coherence & Usability—without using any of the execution or interactive-agent pathways. The exact prompt is provided in Appendix G.4.
>
> **This baseline is trained on the *same data* and follows *the same optimization procedure* as the full GRPO-AR setup (Section 5.2)**. The only difference is that it replaces the agentic reward with a single GPT-5 score based on the raw HTML code, ensuring a clean, controlled comparison. This setup allows us to isolate the contribution of the agentic framework itself, rather than differences in data, reward scaling, or policy optimization.
>
> ---
>
> *Hope our explanation and experiments can address your concerns. We will integrate all your valuable comments into our revision! If you have any further questions or concerns, please feel free to contact us at any time. We are always available and look forward to further discussions with you!*
>
> Best regards,
>
> All Authors

---

> ### Author Response · Authors · 2025-11-27
> **Follow-up on our rebuttal**
>
> Dear Reviewer zdWS,
>
> Wishing you a happy and blessed Thanksgiving!
>
> We wanted to follow up to express our gratitude for your valuable feedback. Over the past few days, we conducted several experiments to address your concerns, such as the **robustness to judge choice** experiment (Appendix H.1). This experiment confirms that our framework works reliably with open-source models, **addressing the reproducibility issues you raised**.
>
> We have updated our manuscript and posted a detailed response to your review. We would be grateful if you could check whether these updates resolve your concerns, and hope you might consider re-evaluating our work in light of these new results and clarifications.
>
> Thanks again for your constructive feedback and looking forward to your reply!
>
> Best regards,
>
> All Authors

---

### Official Review · Reviewer_ZReh · 2025-11-01

**Soundness:** 3
**Presentation:** 3
**Contribution:** 3
**Rating:** 6
**Confidence:** 4

**Summary:**

This paper studies the aesthetic capabilities of code LLMs in visually-oriented coding tasks, such as plot generation and web design, where conventional execution-based metrics are insufficient. To enhance these capabilities, the authors curate a supervised instruction tuning dataset, AesCode358k, covering Python-based plot generation and web design instructions. The model is further improved by reinforcement learning using the GRPO algorithm combined with an Agentic Reward framework, which integrates execution, static aesthetic, and interactive aesthetic evaluations. Finally, the authors propose a new benchmark, OpenDesign, to assess aesthetic quality in LLM-generated web designs.

**Strengths:**

- The paper addresses an underexplored but important problem that is the aesthetic quality of code generated by LLMs for visual tasks, which measuring functional correctness is insufficient .

- The AesCode358k dataset is a valuable contribution that can benefit future research in aesthetic-aware code generation once released.

- The proposed OpenDesign benchmark is also a useful dataset that addresses the limitation of having human voters for evaluation, providing a reproducible way to evaluate aesthetic quality , which enhances scalability and objectivity.

**Weaknesses:**

- The same aesthetic agents are used both during reinforcement learning and evaluation, introducing potential circularity and reward overfitting. The model may learn to exploit or mimic the judge model’s biases rather than improving true generalization in visual aesthetics.

- While improvements on visual coding tasks are reported, the paper lacks an evaluation of whether the aesthetic alignment impacts general code generation ability (e.g., possible regression on standard code benchmarks such as BigCodeBench or LiveCodeBench).

- The Agentic Reward framework relies on multiple large models to serve as static and interactive aesthetic judges. This heavy dependence on high-capacity teacher models will make the framework computationally expensive and difficult to scale, especially for community replication or deployment.

- The definition of “aesthetic quality” remains somewhat opaque. A clearer operationalization or rubric (e.g., color harmony, layout balance, readability) would help readers understand what the models are actually learning to optimize.

**Questions:**

- For the Python-based plot generation data, while executability ensures the code runs without errors, how do the authors verify that the visual output actually matches the instruction semantics (e.g., correct axes, legends, or visual style)?

- In line 169, could the authors elaborate on the clustering method and representative sampling strategy used for data selection?


- In line 175, what are the specific scoring criteria and aspects used by GPT-5 during the aesthetic evaluation?

- How are the relative weights among the three reward components determined? Are they tuned empirically, heuristically, or via grid search?

---

> ### Author Response · Authors · 2025-11-22
> **Response to Reviewer ZReh [1/4]**
>
> We sincerely thank the reviewer for the constructive comments. We address your questions point by point below.
>
> >**W1**: The same aesthetic agents are used both during reinforcement learning and evaluation, introducing potential circularity and reward overfitting. The model may learn to exploit or mimic the judge models biases rather than improving true generalization in visual aesthetics.
>
> We thank the reviewer for raising this concern regarding potential circularity between the reward model used in GRPO-AR training and the evaluation judge. To directly examine whether our results are biased toward the OpenAI judge family, **we re-evaluated the OpenDesign benchmark using `Qwen3-VL-235B-A22B-Instruct`**, a state-of-the-art open-source VLM that is architecturally and stylistically distinct from GPT-4o/GPT-5. The results are reported in Table A.
>
>
>
> **Table A: Results of Mainstream Models on OpenDesign with Different Judges**
> | Model Name | GPT-5 Eval Result| Qwen3-VL-235B-A22B-Instruct Eval Result| Relative Rank|
> | :--- | :---: | :---: |:---:|
> | Claude Sonnet 4 |81.05 |90.90 |1
> | GPT-5 (minimal) |81.03 |90.40 |2
> | Qwen3-Coder-480B-A35B-Instruct |79.90 |90.37 |3
> | DeepSeek-R1-0528 |78.86 |89.50 |4
> | GLM-4.5-Air |78.16 |87.71 |5
> | DeepSeek-V3.1 (Thinking) |77.72 |87.62 |6
> | Qwen3-Coder-30B-A3B-Instruct |73.66 |87.16 |7
> | GLM-4-32B-0414 |69.40 |81.41 |8
> | GPT-4.1 |65.79 |74.46 |9
> | GPT-4o |48.13 |70.52 |10
>
> Across all ten mainstream models, we observe that **the relative ranking remains essentially identical under GPT-5 and Qwen3-VL-235B-A22B-Instruct**, despite notable differences in the absolute scoring scale (Qwen3-VL tends to give higher raw values). This cross-judge consistency demonstrates that **the benchmark’s assessments are not tied to judge-specific preferences of the OpenAI family**, alleviating concerns about reward overfitting.
>
> Moreover, this finding is fully consistent with the human-alignment evidence in Figure 2a. Although the figure reports only GPT-5, our additional experiments show that Qwen3-VL-235B-A22B-Instruct induces the same ordering of the ten mainstream models. Since GPT-5’s ranking already shows strong diagonal agreement with human pairwise preferences, the fact that Qwen3-VL-235B-A22B-Instruct reproduces this ranking provides indirect evidence that **both judges align with human aesthetic preferences in similar ways**.
>
> Taken together, these results indicate that **our model’s improvements are not merely artifacts of exploiting GPT-based reward biases, but reflect genuine gains in aesthetic quality that are preserved across judge families and aligned with human preferences**.
>
> >**W2**: While improvements on visual coding tasks are reported, the paper lacks an evaluation of whether the aesthetic alignment impacts general code generation ability (e.g., possible regression on standard code benchmarks such as BigCodeBench or LiveCodeBench).
>
> Thank you for raising this point. To evaluate whether aesthetic alignment affects standard code generation ability, we additionally measured our models and the base models on three widely used general-purpose code benchmarks: **LiveCodeBench**, **MBPP**, and **MBPP+**. The results are included in Table B.
>
> **Table B: Results of AesCoder and their Base Models on General-Purpose Code Benchmarks**
> | Model | LiveCodeBench[1] | MBPP[2,3] | MBPP+[2,3] |
> | :--- | :---: | :---: |:---:|
> | Qwen3-4B-Instruct-2507 | 32.5 | 86.8 | 74.3 |
> | AesCoder-4B | 19.0 | 73.5 | 64.3 |
> | Qwen2.5-Coder-7B-Instruct | 16.8 | 83.5 | 71.9 |
> | AesCoder-7B | 15.3 | 73.5 | 63.1 |
>
>
> **We observe a slight performance drop compared to the base model, but the regression is small and remains within an acceptable range**. This behavior is expected, as our GRPO-AR training is intentionally specialized for the narrow domain of code aesthetics (e.g. Python-based plot generation, webpage design) rather than broad general code generation ability (e.g. algorithm problems). **Importantly, the model retains strong general code ability and does not exhibit any catastrophic degradation.**
>
> ---
> **References**
>
> [1] Jain, Naman, et al. "Livecodebench: Holistic and contamination free evaluation of large language models for code." arXiv preprint arXiv:2403.07974 (2024).
>
> [2] Austin, Jacob, et al. "Program synthesis with large language models." arXiv preprint arXiv:2108.07732 (2021).
>
> [3] Liu, Jiawei, et al. "Is your code generated by chatgpt really correct? rigorous evaluation of large language models for code generation." Advances in Neural Information Processing Systems 36 (2023): 21558-21572.

---

> > ### Comment · Reviewer_ZReh · 2025-11-25
> > **Response to the authors**
> >
> > Thank you for the detailed and informative responses. They address most of my earlier questions. Regarding the performance impact on general coding tasks, I would gently disagree that the drop is "slight". While I agree this behavior is expected, it would be helpful to explicitly acknowledge this limitation in the paper to provide a more balanced view of the trade-offs. Overall, the clarifications are appreciated.

---

> > > ### Author Response · Authors · 2025-11-26
> > >
> > > We sincerely thank the reviewer for the candid and constructive feedback, and are very glad that your main concerns have been addressed. We fully agree with your assessment that characterizing the performance drop as "slight" was inaccurate. We apologize for this imprecise phrasing in our previous response.
> > >
> > > We acknowledge that this regression is indeed a significant trade-off. As noted by the reviewer ZReh, this is an expected consequence of aligning the model towards a highly specialized domain—code aesthetics—at the expense of broad general-purpose capabilities.
> > >
> > > In response, **we have explicitly highlighted this limitation in Appendix H.3**. We have also included a comprehensive ablation study using an alternative judge in the **Appendix H.1**, clarifying that our model achieves genuine aesthetic improvement rather than simple reward overfitting, and the detailed prompts regarding the scoring criteria mentioned in Line 175 in **Appendix G.8**. All revisions have been highlighted in $\color{blue}{\text{blue}}$ for clarity.
> > >
> > >
> > > We are grateful for your thoughtful feedback and for engaging with our responses throughout the rebuttal process! If you have any further questions or concerns, please feel free to contact us at any time.
> > >
> > > Best regards,
> > >
> > > All Authors

---

> ### Author Response · Authors · 2025-11-22
> **Response to Reviewer ZReh [2/4]**
>
> >**W3**: The Agentic Reward framework relies on multiple large models to serve as static and interactive aesthetic judges. This heavy dependence on high-capacity teacher models will make the framework computationally expensive and difficult to scale, especially for community replication or deployment.
>
> Thank you for raising this concern. We agree that relying on large teacher models (e.g., GPT-5) can increase the computational cost of reward evaluation. However, our analysis shows that the *relative rankings* of models on OpenDesign remain highly stable across different judges, even though their absolute scores may differ (as answered in **W1**). This suggests that the aesthetic reward signal is not tightly coupled to the specific choice of a high-capacity judge, and that **smaller or open-source models can serve as viable substitutes**.
>
> In fact, our experiments with alternative judges indicate that the agentic reward framework maintains its ordering and training behavior, reinforcing the robustness of the approach. **This opens the door to using lighter-weight open-source evaluators as drop-in replacements for GPT-5, substantially reducing the cost for community replication.**
>
> To the best of our knowledge, our work is the first to systematically investigate code aesthetics, and there are no proper open-source aesthetics-oriented reward models for agentic reward framework at present. As part of future work, **we plan to develop fully open-source aesthetic reward models, including both static visual evaluators and interactive agent–based judges, to further reduce the dependence on proprietary systems**. Importantly, the GRPO-AR framework itself is model-agnostic: it can seamlessly switch to any alternative judge model without modifying the optimization procedure, which makes it directly compatible with future open-source replacements.
>
>
> > **W4**: The definition of “aesthetic quality” remains somewhat opaque. A clearer operationalization or rubric (e.g., color harmony, layout balance, readability) would help readers understand what the models are actually learning to optimize.
>
> We appreciate the reviewer’s request for a more explicit operationalization. The "aesthetic quality" is grounded in three concrete and operational criteria used by the static and interactive evaluators. These criteria align with established design principles and directly determine the reward signals observed by the policy.
>
> **(1) Execution-based aesthteic quality.**
> This represents the executability of code. We regard execution-level aesthetic quality as the foundational layer of code aesthetics. Structural validity and executability of code are prerequisites for any meaningful evaluation of higher-level visual or interactive aesthetic properties.
>
>
> **(2) Static aesthetic quality.**
> Static aesthetic quality is assessed based on static images. We define static aesthetic quality as consisting primarily of three dimensions:
>
> - **Color harmony and visual coherence:** consistency of color palettes, avoidance of jarring contrasts, and adherence to recognizable design patterns.
> - **Layout balance and spatial organization:** alignment of elements, whitespace usage, spacing regularity, and avoidance of clutter or collapsed layouts.
> - **Readability and content hierarchy:** clarity of text blocks, font-size coherence, contrast with background, and presence of a reasonable hierarchy of headings and sections.
>
>
> **(3) Interactive aesthetic quality.**
> This dimension measures how well the interactive behaviors support the intended user experience. It evaluates:
>
> - **Functional correctness of widgets:** whether buttons, menus, and links respond as expected;
> - **Meaningful interactions:** avoidance of unnecessary or semantically empty interactions (e.g., gratuitous scrolling or non-functional buttons);
> - **Consistency of interaction patterns:** coherence between layout structure and interactive affordances.
>
> Together, these components provide a clear and operational rubric for aesthetic quality that spans visual, structural, and functional dimensions.

---

> ### Author Response · Authors · 2025-11-22
> **Response to Reviewer ZReh [3/4]**
>
> > **Q1**: For the Python-based plot generation data, while executability ensures the code runs without errors, how do the authors verify that the visual output actually matches the instruction semantics (e.g., correct axes, legends, or visual style)?
>
> **A1**:
>
> Thank you for your valuable question! Inspired by PandasPlotBench[1], we introduce two complementary evaluation methods, each tailored to a different usage scenario:
>
> (1) **With ground-truth plots**. When a ground-truth plot image is available, we provide GPT-5 with the user instruction, the ground-truth image, and the plot generated from the model’s code. GPT-5 leverages its strong visual reasoning capability to compare the two images and assess the discrepancies between the model output and the reference, producing a score for plot fidelity.
>
> (2) **Without ground-truth plots**. When no ground-truth image exists, we supply GPT-5 with the user instruction, all necessary plotting data, and the model-generated plot. GPT-5 then evaluates the plot by comparing it against the instruction and the underlying data, producing a score that reflects adherence to the specified plotting intent.
>
> ---
>  **References**
>
> [1] Galimzyanov, Timur, et al. "Drawing pandas: A benchmark for llms in generating plotting code." 2025 IEEE/ACM 22nd International Conference on Mining Software Repositories (MSR). IEEE, 2025.
>
> ---
>
> > **Q2**:
> In line 169, could the authors elaborate on the clustering method and representative sampling strategy used for data selection?
>
> **A2**:
>
> Thank you for your question. As stated in Appendix B.2, we first embedded all instructions using `openai-text-embedding-3-large` (3072 dimensions). Next, to filter out redundancy, we applied K-Means clustering with $K = 200\text{K}$ on the embedded vectors
> and kept only the sample nearest to each cluster center. This resulted in a refined dataset of 200K
> instructions.
>
> To visualize the effect of our data selection method, we randomly sampled $2000$ instructions from each category and perform t-SNE[1] visualization before and after the filtering process (perplexity = 30, max iter = 1000). See Figure 5 for the t-SNE visualization comparison. The t-SNE visualization of the refined dataset shows clearer class boundaries and
> reduced overlap across categories, demonstrating the effectiveness of our filtering.
>
> ---
> **References**
>
> [1] Maaten, Laurens van der, and Geoffrey Hinton. "Visualizing data using t-SNE." Journal of machine learning research 9.Nov (2008): 2579-2605.
>
> ---
>
> > **Q3**: In line 175, what are the specific scoring criteria and aspects used by GPT-5 during the aesthetic evaluation?
>
> **A3**:
>
> We apologize for the unclear wording here. The scoring mechanism used in Line 175 with GPT-5 mainly contains three dimensions:
>
> - **Alignment with User Instruction**.
> Measures how completely and precisely the generated webpage fulfills both explicit and implicit requirements of the user prompt, including the presence, correctness, and faithful implementation of all requested elements and overall correspondence between content, structure, and the instruction.
>
> - **Visual Aesthetics and Readability**.
> Evaluates the visual appeal and professionalism of the webpage, including the consistency and effectiveness of color, typography, and spacing choices, as well as the clarity, readability, and overall visual polish of the layout.
>
> - **Structural Integrity and Cohesion**.
> Assesses whether the webpage’s structure is logical, well-organized, and cohesive, with smooth section-to-section flow and an intuitive user experience.
>
> The full prompt is in Appendix G.8.

---

> ### Author Response · Authors · 2025-11-22
> **Response to Reviewer ZReh [4/4]**
>
> >**Q4**: How are the relative weights among the three reward components determined? Are they tuned empirically, heuristically, or via grid search?
>
> **A4**:
>
> The weight configuration was selected through **a coarse-to-fine exploratory process** designed to understand the qualitative contribution of each reward component rather than through an expensive full-grid sweep.
>
> Since modern web browsers are highly tolerant of minor syntactic imperfections in HTML, we fixed $w_{\text{exec}} = 0.1$ to ensure a minimum level of executability and redistributed the remaining mass between the static and interactive terms. **As an initial diagnostic, we examined two extremes**: a *static-only* setting $(w_{\text{static}}=0.9, w_{\text{interact}}=0)$ and an *interactive-only* setting $(w_{\text{static}}=0, w_{\text{interact}}=0.9)$. These endpoints revealed complementary failure modes: the static-dominant configuration yielded visually coherent but functionally limited pages, whereas the interaction-dominant configuration tended to over-optimize for actionability, introducing unnecessary or semantically weak interactive elements and noticeably degrading visual aesthetics.
>
> Guided by these observations, **we then performed a lightweight grid exploration** by varying $w_{\text{static}}$ in increments of $0.2$ while keeping the total mass fixed. For each configuration, we inspected rollout behaviors and monitored both static- and interaction-based validation signals. The final setting $(w_{\text{exec}}, w_{\text{static}}, w_{\text{interact}}) = (0.1, 0.8, 0.1)$ provided the most balanced trade-off: it preserved high static aesthetic quality while avoiding the over-activation behaviors observed in more interactive-heavy regimes. We therefore used this configuration for the main GRPO-AR experiments.
>
>
> ---
>
> *Hope our explanation and experiments can address your concerns. We will integrate all your valuable comments into our revision! If you have any further questions or concerns, please feel free to contact us at any time. We are always available and look forward to further discussions with you!*
>
> Best regards,
>
> All Authors

---

### Official Review · Reviewer_k5dx · 2025-11-01

**Soundness:** 3
**Presentation:** 3
**Contribution:** 2
**Rating:** 6
**Confidence:** 3

**Summary:**

The paper proposes a training and evaluation pipeline to improve the visual quality ("code aesthetics") of LLM generated code in two domains: Python plotting and webpage design. The authors (i) build AesCode-358K, a supervised instruction tuning dataset comprising approximately 158k validated plotting examples and a large set of webpage design cases produced and filtered through a multi step process, (ii) introduce an agentic reward framework with three agents for execution, static aesthetics, and interactive aesthetics, and aggregate their signals with a weighted sum, (iii) train AesCoder-4B/7B with GRPO-AR, and (iv) propose OpenDesign, a benchmark for static and interactive webpage aesthetics whose rankings show high agreement with the Design Arena leaderboard and reasonable GPT and human alignment. On PandasPlotBench and OpenDesign, AesCoder-4B outperforms GPT-4o/4.1 and is competitive with much larger models, and ablation studies show benefits over DPO and RFT and over GRPO without the agentic reward.

**Strengths:**

The problem is well defined, the methodology is mostly sound, and the experiments support the main claims. The reliance on proprietary judges (GPT-5/GPT-4o) for both data curation and evaluation introduces possible bias, which the paper partially mitigates via correlation with Design Arena and human annotations. More analysis on sensitivity to judge choice and reward weights would strengthen soundness.

The execution agent uses HTMLHint rules rather than brittle strict parsing; the static aesthetics agent defines three explicit criteria and uses a single image of the rendered page; the interactive agent uses a GUI agent to act on the page and aggregates successes. These design choices are concrete and implementable.

OpenDesign’s rankings have high correlation with Design Arena (Spearman 0.98, Kendall 0.91), and GPT–human agreement of 80.9% exceeds human–human agreement for the 200 pairwise comparisons, which supports the reliability of the evaluation protocol.

**Weaknesses:**

GPT-5 (and GPT-4o) are used during dataset filtering, for static aesthetics scoring, and within the interactive agent. This may bias the training signal and the benchmark toward these judges’ preferences. While OpenDesign shows strong rank correlation with Design Arena and decent GPT–human agreement, results could be sensitive to the choice of judge. Please report results with at least one strong alternative judge and quantify changes.

The paper defines the weighted sum for r and also reports a concrete setting of the weights ($w_{exec}$, $w_{static}$, $w_{interact}$). However, it does not describe a normalization scheme across agents or provide an ablation over alternative weight choices. This limits reproducibility and makes it hard to assess sensitivity. Please report the exact normalization used and add a systematic ablation over plausible ranges, for example, varying one weight while keeping the total fixed, and include confidence intervals for all metrics.


A few typos reduce polish (e.g., “sucess”, “Grammer”, “protion”). Figures are informative but Figure 1 would benefit from a clearer depiction of when each agent runs (serial vs parallel) and where screenshots are taken.

**Questions:**

What are the exact definitions and numeric ranges of the three agent signals before aggregation, and is any normalization, clipping, or rescaling applied to each signal prior to the weighted sum?

The paper reports one specific setting of the weights w_exec, w_static, and w_interact. Were these exact weights used for all training runs and all evaluations, and how were they chosen in the first place for example by a validation set criterion or a simple search?

How sensitive are the results to the choice of weights? Please provide a systematic ablation that varies the weights over a grid while keeping the total weight fixed, and report primary metrics with confidence intervals and the stability of model rankings.

How should users of the method select weights in new domains or tasks that differ from plots and webpages? Please provide general guidance or a simple procedure for choosing weights based on observable properties of the three signals, and discuss when rebalancing the weights is necessary.

---

> ### Author Response · Authors · 2025-11-22
> **Response to Reviewer k5dx [1/3]**
>
> We sincerely thank the reviewer for the constructive comments. We address your questions point by point below.
>
> >**W1**:
> While OpenDesign shows strong rank correlation with Design Arena and decent GPT–human agreement, results could be sensitive to the choice of judge. Please report results with at least one strong alternative judge and quantify changes.
>
> We thank the reviewer for highlighting the potential bias introduced by using GPT-4o/GPT-5 as the primary judges. To address this, **we conducted a new evaluation on the OpenDesign benchmark using `Qwen3-VL-235B-A22B-Instruct` as the judge with the same scoring prompt**. We selected this model because it is a state-of-the-art open-source Vision-Language Model with strong reasoning capabilities, distinct from the OpenAI family. The results are shown in Table A.
>
>
> **Table A: Results of Mainstream Models on OpenDesign with Different Judges**
> | Model Name | GPT-5 Eval Result| Qwen3-VL-235B-A22B-Instruct Eval Result| Relative Rank|
> | :--- | :---: | :---: |:---:|
> | Claude Sonnet 4 |81.05 |90.90 |1
> | GPT-5 (minimal) |81.03 |90.40 |2
> | Qwen3-Coder-480B-A35B-Instruct |79.90 |90.37 |3
> | DeepSeek-R1-0528 |78.86 |89.50 |4
> | GLM-4.5-Air |78.16 |87.71 |5
> | DeepSeek-V3.1 (Thinking) |77.72 |87.62 |6
> | Qwen3-Coder-30B-A3B-Instruct |73.66 |87.16 |7
> | GLM-4-32B-0414 |69.40 |81.41 |8
> | GPT-4.1 |65.79 |74.46 |9
> | GPT-4o |48.13 |70.52 |10
>
> The results in Table A clearly show that, despite differences in absolute score scales—Qwen3-VL-235B-A22B-Instruct typically assigns higher numerical values under the same scoring prompt—**the relative ordering of all mainstream models remains highly consistent between GPT-5 and Qwen3-VL-235B-A22B-Instruct**. This strong rank alignment indicates that OpenDesign is **robust to the choice of judge** and that **our reported improvements are not artifacts of overfitting to GPT-based evaluators, but rather reflect genuine gains in design quality**.
>
> Moreover, this rank consistency is aligned with the human-preference results reported in Figure 2a. Although the figure shows GPT-5 as the judge, our additional experiments demonstrate that **Qwen3-VL-235B-A22B-Instruct produces essentially the same relative ranking as GPT-5 across the 10 mainstream models**. Since GPT-5’s ranking already shows strong diagonal agreement with human pairwise preferences, the fact that Qwen3-VL-235B-A22B-Instruct replicates this ranking structure provides indirect but compelling evidence that **Qwen3-VL-235B-A22B-Instruct is similarly aligned with human aesthetic preferences**. In other words, both judges—despite belonging to different model families—capture the same underlying human-preferred ordering on OpenDesign.
>
>
> >**W2**: The paper defines the weighted sum for $x$ and also reports a concrete setting of the weights ($w_{exec}, w_{static}, w_{interact}$). However, it does not describe a normalization scheme across agents or provide an ablation over alternative weight choices. This limits reproducibility and makes it hard to assess sensitivity. Please report the exact normalization used and add a systematic ablation over plausible ranges, for example, varying one weight while keeping the total fixed, and include confidence intervals for all metrics.
>
> For normalization details, please see **A1** below. For the ablation study, please see **A3** below.
>
>
> >**W3**: A few typos reduce polish (e.g., "success", "Grammer", "protion"). Figures are informative but Figure 1 would benefit from a clearer depiction of when each agent runs (serial vs parallel) and where screenshots are taken.
>
> Thank you for highlighting these issues! We apologize for the typos and the lack of clarity in Figure 1. We have fixed the typos and updated Figure 1 in the revised version.

---

> ### Author Response · Authors · 2025-11-22
> **Response to Reviewer k5dx [2/3]**
>
> > **Q1**: What are the exact definitions and numeric ranges of the three agent signals before aggregation, and is any normalization, clipping, or rescaling applied to each signal prior to the weighted sum?
>
> **A1**:
> 1. **The Exact Definitions and Numeric Ranges of the Three Agent Signals**
>
>     - **Execution Agent Signal**: It represents the code execution check (e.g. the HTMLHint check for HTML) result of the model's output. It has two value. $1$ if the output passes all validations, or $-1$ otherwise.
>     - **Static Aesthetics Agent Signal**: It measures the static aesthetics quality of the webpage. It is scored by an LLM-judger only based on the static screenshot of the webpage and mainly focuses on these three dimensions: `Alignment with User Instruction`, `Element Aesthetics and Readability`, and `Structural Integrity and Cohesion`. It is an integer between $0$ and $100$.
>     - **Interactive Aesthetics Agent Signal**: It measures the functionality of the interactive webpage elements, for example, buttons and text boxes. It is the number of successful interactions, so it is an integer between $0$ and `max_iter`. In our experiments, `max_iter` is set to $3$.
>
> 2. **Normalization and Rescaling to Each Signal**
>
>     Suppose $r$ is the term in weighted sum transformed from the origin signal, $s$ is the origin signal.
>
>
>     - Execution Agent Signal: $r_{exec}=s_{exec}$.
>     - Static Aesthetics Agent Signal: $r_{static}=(s_{static}-50)/50$.
>     - Interactive Aesthetics Agent Signal: $r_{interact}=s_{interact}-1$.
>
>     Because of the gating mechanism we employ, **where the static and interactive aesthetics agents are activated only after the executability check succeeds**,
>     and the final reward aggregation equation is:
> $$
> r = \\begin{cases}
>     r_{\text{exec}}, & r_{\text{exec}} = -1 \\\\
>     w_{\text{exec}} \cdot r_{\text{exec}} + w_{\text{static}} \cdot r_{\text{static}} + w_{\text{interact}} \cdot r_{\text{interact}}, & r_{\text{exec}} = 1
> \\end{cases}
> $$
>     where $w$ represents the weight assigned to each agent.
>
>
> > **Q2**: The paper reports one specific setting of the weights w_exec, w_static, and w_interact. Were these exact weights used for all training runs and all evaluations, and how were they chosen in the first place for example by a validation set criterion or a simple search?
>
> **A2**:
>
> (1) Yes. These exact weights are used for all training runs and all evaluations.
>
> (2) The weight configuration was selected through **a coarse-to-fine exploratory process** designed to understand the qualitative contribution of each reward component rather than through an expensive full-grid sweep.
>
> Since modern web browsers are highly tolerant of minor syntactic imperfections in HTML, we fixed $w_{\\text{exec}} = 0.1$ to ensure a minimum level of executability and redistributed the remaining mass between the static and interactive terms. **As an initial diagnostic, we examined two extremes**: a *static-only* setting $(w_{\\text{static}}=0.9,\\; w_{\\text{interact}}=0)$ and an *interactive-only* setting $(w_{\\text{static}}=0,\\; w_{\\text{interact}}=0.9)$. These endpoints revealed complementary failure modes: the static-dominant configuration yielded visually coherent but functionally limited pages, whereas the interaction-dominant configuration tended to over-optimize for actionability, introducing unnecessary or semantically weak interactive elements and noticeably degrading visual aesthetics.
>
> Guided by these observations, **we then performed a lightweight grid exploration** by varying $w_{\text{static}}$ in increments of $0.2$ while keeping the total mass fixed. For each configuration, we inspected rollout behaviors and monitored both static- and interaction-based validation signals. The final setting $(w_{\\text{exec}}, w_{\\text{static}}, w_{\\text{interact}}) = (0.1, 0.8, 0.1)$ provided the most balanced trade-off: it preserved high static aesthetic quality while avoiding the over-activation behaviors observed in more interactive-heavy regimes. We therefore used this configuration for the main GRPO-AR experiments.

---

> ### Author Response · Authors · 2025-11-22
> **Response to Reviewer k5dx [3/3]**
>
> > **Q3**: How sensitive are the results to the choice of weights? Please provide a systematic ablation that varies the weights over a grid while keeping the total weight fixed, and report primary metrics with confidence intervals and the stability of model rankings.
>
> **A3**:
>
> Thank you for the suggestion. To assess the sensitivity of our agentic reward framework to weight choices, we fixed $w_{\text{exec}} = 0.1$ and evaluated **five representative configurations** by training GRPO-AR on $\text{AesCoder-4B}_\text{sft}$ under each setting. *Due to computational constraints*, we ran **150 RL steps** per configuration and evaluated each resulting model on OpenDesign **five times** to compute 95% confidence intervals. The reward curves are included in Appendix H.2.
>
> **Table B: Results on OpenDesign Under Different Weight Choices**
>
> | $w_{exec}$ | $w_{static}$ | $w_{interact}$ | Static Aesthetic Score     | Interactive Aesthetic Score     |
> |:------------:|:--------------:|:----------------:|:-----------------------------:|:---------------------------------:|
> | $0.1$      | $0.0$        | $0.9$          | $78.88 \pm 0.32$            | $0.91 \pm 0.20$                 |
> | $0.1$      | $0.2$        | $0.7$          | $79.28 \pm 0.30$            | $0.87 \pm 0.17$                 |
> | $0.1$      | $0.4$        | $0.5$          | $79.87 \pm 0.33$            | $0.84 \pm 0.18$                 |
> | $0.1$      | $0.6$        | $0.3$          | $80.01 \pm 0.33$            | $0.80 \pm 0.17$                 |
> | $0.1$      | $0.9$        | $0.0$          | $80.50 \pm 0.32$            | $0.63 \pm 0.15$                 |
>
> Across settings, we observe a **smooth and interpretable trade-off**: increasing $w_{\text{static}}$ improves static aesthetic quality, while decreasing $w_{\text{interact}}$ naturally reduces interactive performance, and vice versa. This behavior is consistent with the intended reward decomposition, indicating that the framework responds predictably to changes in weight allocation rather than exhibiting brittle or unstable dynamics.
>
> Importantly, the **reward curves remain stable and follow a normal upward trajectory** under all tested configurations, with no signs of divergence or collapse. This suggests that GRPO-AR maintains stable optimization dynamics across a wide range of weight settings and that the agentic reward components provide **reliable and steerable training signals**.
>
> > **Q4**:
> How should users of the method select weights in new domains or tasks that differ from plots and webpages? Please provide general guidance or a simple procedure for choosing weights based on observable properties of the three signals, and discuss when rebalancing the weights is necessary.
>
> **A4**:
>
> >**Guidance on selecting weights in new domains.**
>
> Our method does not rely on domain-specific heuristics; instead, the three reward components exhibit consistent and interpretable behavior across tasks. For new domains, we recommend a simple and robust two-stage procedure.
>
> **(1) Determine $w_{exec}$ based on Environmental Fault Tolerance.**
>
> First, fix the execution weight based on the strictness of the target language's syntax.
>
> - For high-tolerance domains (e.g., HTML/CSS rendering), we recommend a low, fixed weight (e.g., $w_{exec}=0.1$). As web browsers are permissive, a high penalty here is unnecessary and may hinder exploration.
> - For low-tolerance domains (e.g., Python/C++ execution), where a single syntax error causes total failure, $w_{exec}$ must be set significantly higher to serve as a hard constraint.
>
>
> **(2) Calibrate $w_{static}$ and $w_{interact}$ via Coarse Grid Search and Qualitative Verification**
>
> Once $w_{exec}$ is fixed, we recommend performing a coarse grid search (e.g., steps of 0.2) for the remaining weights. Crucially, selection should not rely solely on reward curves but on qualitative verification of intermediate checkpoints.
>
> > **Cases when rebalancing weights becomes necessary.**
>
> Rebalancing is required only when the new domain exhibits structural trade-offs different from plots or webpages. Typical signals include:
> - high visual fidelity but missing or unreliable interactions: increase $w_{\text{interact}}$;
> - excessive, unnecessary, or semantically meaningless interactivity: decrease $w_{\text{interact}}$, increase $w_{\text{static}}$;
> - frequent execution failures: slightly increase $w_{\text{exec}}$;
> - saturation of any reward component early in training: renormalize or adjust its weight.
>
> This procedure worked consistently in both plotting and webpage design. Importantly, the coarse sweep requires only a small number of rollouts and does not require full RL retraining for each candidate.
>
> ---
>
> *Hope our explanation and experiments can address your concerns. If you have any further questions or concerns, please feel free to contact us at any time. We are always available and look forward to further discussions with you!*
>
> Best regards,
>
> All Authors

---

### Author Response · Authors · 2025-11-26
**Rebuttal Summary: New Experiments & Clarifications**

Dear Reviewers and ACs,

We sincerely thank you for your time and constructive feedback. We have carefully considered all comments and updated the paper, with changes highlighted in $\color{blue}{\text{blue}}$.

During the rebuttal period, we conducted extensive additional experiments and provided detailed clarifications to address reviewers' questions. These additions validate and strengthen our original conclusions:

### **Additional Experiments & Revisions**
* **Robustness to Judge Choice (k5dx, ZReh, zdWS, Z31k):** To address concerns about proprietary model bias, we re-evaluated OpenDesign using the open-source **Qwen3-VL-235B-A22B-Instruct** model as an alternative judge ***(Appendix H.1)***. **Result:** Relative rankings remain highly stable, confirming that our improvements reflect genuine aesthetic gains rather than overfitting to GPT preferences.
* **Weight Ablation & Stability (k5dx, zdWS, Z31k):** We conducted a systematic grid search varying reward weights ***(Appendix H.2)***. **Result:** Reward curves show stable convergence. We clarified that GRPO's group relative advantage calculation creates an implicit curriculum that ensures synergistic optimization without instability.
* **Impact on General Code Generation (ZReh):** We evaluated AesCoder on **LiveCodeBench**, **MBPP** and **MBPP+** ***(Appendix H.3)***. **Result:** While aesthetic specialization incurs a moderate and expected regression, the model retains functional competence without catastrophic degradation.

### **Key Clarifications**
* **Computational Feasibility (zdWS):** We clarified that the pipeline is **CPU-bound** (rendering/execution) and lightweight (~25s/sample). With open-source judges now validated, the framework is fully accessible to academic labs.
* **Failure Case Study (Z31k):** We added **failure case studies** ***(Appendix I)*** demonstrating how our cascading reward mechanism effectively penalizes deceptive outputs (e.g., visually correct but non-executable code).
* **Scoring Criteria (ZReh):** Detailed prompts for scoring criteria in Line 175 have been added to ***Appendix G.8***.
* **Generalization to New Domains (zdWS, Z31k):** We clarified that OpenDesign already encompasses diverse tasks like **Game Development** and **UI Components**, confirming that the framework is generalizable across diverse domains beyond simple webpages.

---

We thank all reviewers for their constructive feedback, which has substantially strengthened our work. We believe these extensive revisions thoroughly address all concerns raised and demonstrate the solid contributions of AesCoder to visually-oriented code generation.

Best regards,

All Authors

---

### Author Response · Authors · 2025-12-03
**Rebuttal Summary for AC**

Dear AC,

In light of the unusual rebuttal process, we provide this concise summary to assist your assessment. **All points below are grounded directly in the discussion thread.**
### **Key Strengths Recognized by Reviewers**
We are encouraged that all reviewers consistently recognized the core value of our work:
- **Pioneering Problem & Novelty (Reviewers zdWS, ZReh):** Highlighted as "the first systematic effort to formalize code aesthetics" and addresses an "underexplored but important problem."
- **Sound Methodology & Framework (Reviewers zdWS, k5dx):** The Agentic Reward framework was praised as a "novel architecture" that "synergistically combines textual analysis, visual rendering, and interactive evaluation."
- **High-Quality Resources (Reviewers Z31k, ZReh):** The AesCode-358K dataset and OpenDesign benchmark were recognized as "valuable contributions" that are "reproducible and scalable."
- **Strong Results (Reviewer k5dx):** The state-of-the-art performance and its "strong rank correlation" with human preferences were explicitly acknowledged.
### **Summary of Reviewer Responses**
Prior to the system freeze, our submission received explicit positive endorsements from *all* the reviewers who engaged in the discussion.

* **Z31k (Score: 4 $\to$ 6):** Explicitly stated that *“The authors have clarified the key concerns... I am raised my score.”* (Comment dated 26 Nov 2025, 03:19 UTC, before the leak incident)
* **ZReh (Score: 6 $\to$ 6):** Confirmed that *“the authors have addressed most of earlier concerns”* and maintained the positive rating. (Comment dated 25 Nov 2025, 21:07 UTC, before the leak incident)

For the two reviewers who did not leave final comments (**k5dx, Initial: 6 and zdWS, Initial: 4**), their primary concerns (Judge Bias, Generalization, Reproducibility) were **identical** to those raised by Z31k and ZReh. Since our new experiments (e.g., Qwen3-VL cross-validation, weight ablation) successfully resolved these exact issues for the active reviewers, **we are confident the manuscript now meets the acceptance bar for all.**
### **Detailed Breakdown**
| Reviewer | Key Strengths | Concerns/Questions (All Resolved) | Pre/Post-Rebuttal Rating |
| :--- | :--- | :--- |:---|
| **k5dx** | [1] Well-defined problem with sound methodology and concrete agent design. [2] Reliable benchmark (OpenDesign) highly correlated with human preference. | [1] **Judge Bias:** Reliance on proprietary models for data/evaluation. [2] **Robustness:** Lack of weight ablation and normalization details. [3] **Presentation:** Minor typos and clarity issues. | **- Pre: 6 (Confidence: 3) - Post**: No final comment. However, given that we conducted thorough experiments regarding **judge bias & weight ablation (Appendix H.1, H.2)**, and **addressed the clarification questions in detail**, we believe there is a strong chance the score would increase. |
| **ZReh** | [1] Important, underexplored problem setting. [2] Valuable contributions of the AesCode-358K dataset and reproducible benchmark. | [1] **Judge Bias & Cost:** Potential reward overfitting and high computational expense. [2] **General Ability:** Lack of evaluation on general code generation tasks. [3] **Definitions:** "Aesthetic quality" definition appeared opaque. | **- Pre: 6 (Confidence: 4) - Post**: Reviewer stated that **most of the questions have been addressed** and **kept positive for our paper on Nov 26**.  |
| **zdWS** | [1] First systematic, rigorous effort to formalize code aesthetics. [2] High significance for real-world deployment and scalability. | [1] **Reproducibility:** Barriers due to proprietary models and computational cost. [2] **Generalization:** Scope limited to plots/webpages; untested on Games/GUIs. [3] **Robustness:** Insufficient detail on interactive agent failure modes. | **- Pre: 4 (Confidence: 3) - Post**: No final comment. However, the primary concern on Judge Bias/Reproducibility **aligns with Z31k**, who explicitly confirmed resolution and raised score. All remaining concerns were clarifications that we fully addressed. Therefore, **we believe the resolution of these concerns warrants an increased rating, likely 6**.|
| **Z31k** | [1] Substantial contributions in an underexplored area. [2] Improvements validated by human preference tests. | [1] **Generalization:** Applicability to other modalities. [2] **Robustness:** Handling ambiguous outputs and weight selection sensitivity. [3] **Judge Bias:** Potential overfitting to automated metrics. | **- Pre: 4 (Confidence: 3) - Post**: Reviewer stated on **26 Nov 2025**: "Thank you for the rebuttal. The authors have clarified the key concerns I raised. I am raised my score." **The reviewer subsequently raised the score to 6.** |

We provide a detailed summary of all additional experiments and clarifications in comment "**Rebuttal Summary: New Experiments & Clarifications**" below.

We appreciate your time and consideration in reviewing our work during this unusual rebuttal process.

---

### Meta-Review · Area_Chair_kEz6 · 2026-01-06

**Summary:**

This paper proposes a novel benchmark for code generation in domains where the code generates a visual artifact (e.g., HTML or plotting code), and we want to optimize the aesthetics of this artifact. The authors also propose a finetuning methodology for training a model that outperforms closed models.

**Reviewer Concerns:**

The reviewers raised a number of concerns, the most notable ones being: (1) the heavy reliance on LLM judges, which may be mis-aligned, (2) the complexity and stability of the post-training procedure, and (3) the scope of domains considered. There was substantial progress to addressing all of these concerns.

The most notable remaining concern I have is related to (1), namely, that even if an LLM-judge is aligned (as the authors show), *optimizing* an input against that LLM-judge can exploit weaknesses in its capabilities to mis-align it. This is similar to how adversarial examples work. While the authors demonstrate that their results remain consistent according to a new judge (a Qwen model), I did not see an evaluation of the finetuned model, AesCoder-4B, using Qwen (though I may have missed it), and furthermore, it is well known that adversarial examples are transferrable across models. Thus, I believe that this issue has been significantly mitigated, though still remains in some form. To further increase confidence in the results, I would strongly encourage the authors to include human alignment results not just on the ranking of blackbox models, but also on the outputs of AesCoder-4B.

Finally, I note that the reviewers generally agreed that this paper is tackling an important problem and has a significant contribution.

**Reviewer Scores:**

One reviewer messaged that they increased their score during the rebuttal period (this reviewer is the one who pointed out the issue regarding overfitting the LLM-Judge). The other negative reviewer did not respond, but I believe their concerns were addressed to a large degree.

---

### Decision · Program_Chairs · 2026-01-26

Accept (Poster)